# Mitigating Simplicity Bias in Deep Learning for Improved OOD Generalization and Robustness

**Bhavya Vasudeva**                                                           *bvasudev@usc.edu*
*University of Southern California*

**Kameron Shahabi**                                                          *kyshahab@usc.edu*
*University of Southern California*

**Vatsal Sharan**                                                            *vsharan@usc.edu*
*University of Southern California*

**Reviewed on OpenReview:** *https://openreview.net/forum?id=XccFHGakyU*

## Abstract

Neural networks (NNs) are known to exhibit *simplicity bias* where they tend to prefer learning 'simple' features over more 'complex' ones, even when the latter may be more informative. Simplicity bias can lead to the model making biased predictions which have poor out-of-distribution (OOD) generalization and subgroup robustness. To address this, we propose a hypothesis about spurious features that directly connects to simplicity bias: we hypothesize that spurious features on many datasets are simple features that are still predictive of the label. We empirically validate this hypothesis, and subsequently develop a framework which leverages this hypothesis to learn more robust models. In our proposed framework, we first train a simple model, and then regularize the conditional mutual information with respect to it to obtain the final model. We theoretically study the effect of this regularization and show that it provably reduces reliance on spurious features in certain settings. We also empirically demonstrate the effectiveness of this framework in various problem settings and real-world applications, showing that it effectively addresses simplicity bias and leads to more features being used, enhances OOD generalization, and improves subgroup robustness and fairness.

## 1 Introduction

Motivated by considerations of understanding generalization in deep learning, there has been a series of interesting studies (Zhang et al., 2017a; Frankle & Carbin, 2019; Nakkiran et al., 2020) on understanding function classes favored by current techniques for training large neural networks. An emerging hypothesis is that deep learning techniques prefer to learn simple functions over the data. While this inductive bias has benefits in terms of preventing overfitting and improving (in-distribution) generalization in several cases, it is not effective in all scenarios. Specifically, it has been found that in the presence of multiple predictive features of varying complexity, neural networks tend to be overly reliant on simpler features while ignoring more complex features that may be equally or more informative of the target (Shah et al., 2020; Nakkiran et al., 2019; Morwani et al., 2023). This phenomenon has been termed *simplicity bias* and has several undesirable implications for robustness and out-of-distribution (OOD) generalization.

As an illustrative example, consider the Waterbirds dataset (Sagawa* et al., 2020). The objective here is to predict a bird's type (landbird vs. waterbird) based on its image (see Fig. 1 for an example). While features such as the background (land vs. water) are easier to learn, and can have a significant correlation with the bird's type, more complex features like the bird's shape are more predictive of its type. However, simplicity bias can cause the model to be highly dependent on simpler yet predictive features, such as the background in this case. A model which puts high emphasis on the background for this task is not desirable, since its performance may not transfer across different environments. A similar story arises in

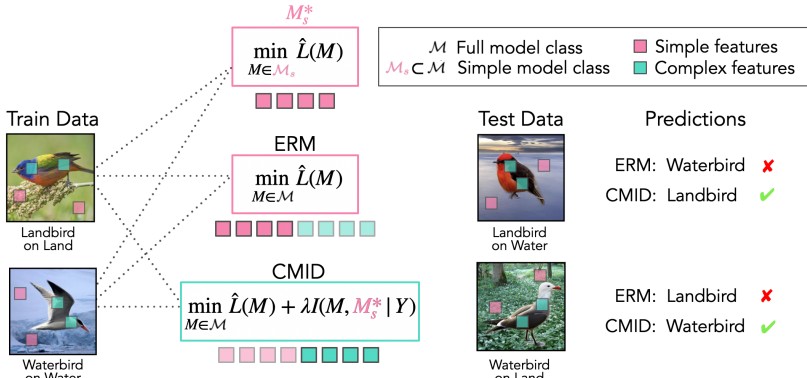

Figure 1: Summary of our approach. Models trained with ERM tend to focus on simple features (such as background) that may not generalize OOD, whereas encouraging conditional independence with respect to a simple model increases reliance on complex features (such as shape) that generalize.

| Dataset | Task-relevant/invariant feature | Surrogate/spurious feature |
|---------|---------------------------------|----------------------------|
| Waterbirds | Bird type (waterbird/landbird) | Background (water/land) |
| CelebA | Hair colour (blonde/other) | Gender (female/male) |
| MultiNLI | Reasoning (entailment/neutral/contradiction) | Negation words ('not' etc.) |
| CivilComments-WILDS | Sentiment (toxic/non-toxic) | Demographic attributes (race, gender, religion) |
| Colored-MNIST | Digit ($< 5$ or $\geq 5$) | Color (red/green) |
| Camelyon17-WILDS | Diagnoses (tumor/no tumor) | Hospital |
| Adult-Confounded | Income ($< \$50k$ or $\geq \$50k$) | Demographic attributes (race, gender) |

Table 1: Summary of the datasets we consider. Spurious features seem *simpler* than invariant features.

many different tasks—Table 1 summarizes various datasets, where the target or task-relevant features (also known as invariant features), are more complex than surrogate features that are superficially correlated with the label (also known as spurious features). Simplicity bias causes NNs to heavily rely on these surrogate features.

Several methods (Arjovsky et al., 2020; Creager et al., 2021; Bae et al., 2022; Gao et al., 2022) have been proposed to address this problem of OOD generalization. However, some knowledge of the different environments of interests or the underlying causal graph is in general necessary to decide whether a feature is spurious or invariant (due to the No Free Lunch Theorem, see Appendix D for more discussion). Therefore, most techniques in the literature require this information about whether a feature is spurious or invariant to be available beforehand. To develop a principled framework for OOD generalization that does not use such prior knowledge, we believe it is useful to first start with a suitable assumption of what features will be regarded as being spurious. In this work, we use simplicity bias and observations from Table 1 to propose such a candidate hypothesis about spurious features: *Features typically regarded as being spurious are relatively simple while still being predictive of the label.* If this hypothesis is true, alleviating simplicity bias can lead to better robustness and OOD generalization.

We empirically evaluate this hypothesis and find it to be valid on considered datasets. In light of these results, we develop a new framework to mitigate simplicity bias and encourage the model to utilize a more diverse set of features for its predictions, and hence improve robustness and OOD generalization. The high-level goal of our framework is to ensure that the predictions of the trained model $M$ have minimal conditional mutual information $I(M; S|Y)$[1] with any simple, predictive feature $S$, conditioned on the label $Y$. To achieve this, we first train a simple model $M_s^*$ on the task, with the idea that this model captures the information in simple, predictive features. Subsequently, to train the final model $M$ we add its conditional mutual information $I(M; M_s^*|Y)$ with the model $M_s^*$ as a regularizer to the usual empirical risk minimization (ERM) objective. With this regularization term, we incentivize the model to leverage additional, task-relevant features that

---

[1]With slight notation abuse, $M$ denotes both the model and the random variable associated with its predictions.

may be more complex. We refer to our approach as *Conditional Mutual Information Debiasing or CMID* (see Fig. 1).

We show that our approach leads to models that use more diverse features on tasks that have been previously proposed to measure simplicity bias. The method achieves improved subgroup robustness and OOD generalization on several benchmark tasks, including in the presence of multiple spurious features. It also leads to improved predictions from the perspective of fairness since the predictions are less dependent on protected attributes such as race or gender. We also prove theoretical results to better understand and characterize our approach. Our contributions are:

- We empirically evaluate and validate the hypothesis that spurious features are simpler than invariant features, with respect to representative datasets from Table 1 (Section 2). Based on our findings, we propose a novel framework (CMID) to mitigate a model's simplicity bias and use a diverse set of features for prediction (Section 3).

- We theoretically analyze the effect of our regularization and demonstrate its capability to reduce reliance on spurious features using a Gaussian mixture model setup (Section 4). Additionally, we establish an OOD generalization guarantee in a causal learning framework (Section 4). Most approaches developed for subgroup robustness or OOD generalization either lack theoretical guarantees or show results tailored to one of these two settings (discussed further in Appendix C).

- Empirically, we demonstrate that our framework effectively mitigates simplicity bias, and achieves improved OOD generalization, sub-group robustness and fairness across 10 benchmark datasets (Section 5). These include different modalities such as image, text, and tabular data and application domains such as healthcare and fairness. While several approaches have been proposed for each of the goals we consider, prior work typically focuses on one or two of these applications, while our approach proves effective across all cases.

## 1.1 Related Work

We discuss related work on simplicity bias, OOD generalization and subgroup robustness in this section, and related work on fairness, feature diversification and debiasing methods in Appendix B. A detailed comparison with prior work appears in Appendix C, here we summarize some key differences. A conceptual difference from prior algorithms for these tasks is that we take a more data-driven approach to define spurious features. Hence our approach is more explicit in terms of its assumptions on the data which could make it easier to evaluate and use, and also allows us to prove theoretical guarantees. Various datasets seem to satisfy our assumptions and hence our approach proves effective across varied applications. Moreover, in contrast to other methods, it does not involve training multiple complex models, access to group labels, or unlabelled data from the target distribution.

**Simplicity Bias in NNs.** Several works (Arpit et al., 2017; Valle-Perez et al., 2019; Geirhos et al., 2020; Pezeshki et al., 2021) show that NNs trained with gradient-based methods prefer learning solutions which are 'simple'. Nakkiran et al. (2019) show that the predictions of NNs trained by SGD can be approximated well by linear models during the early stages of training. Morwani et al. (2023) show that 1-hidden layer NNs are biased towards using low-dimensional projections of the data for predictions. Geirhos et al. (2022) show that CNNs make predictions that depend more on image texture than image shape. Shah et al. (2020) create synthetic datasets and show that in the presence of simple and complex features, NNs rely heavily on simple features even when both have equal predictive power. In our work, we experiment with datasets in both these papers (in addition to the datasets in Table 1), to investigate the effectiveness of our method to mitigate simplicity bias.

**OOD Generalization.** Towards developing models that perform better in the real world, OOD generalization requires generalization to data from new *environments*. Environments are usually defined based on the correlation between some spurious feature and the label. Various methods aim to recover a predictor that is *invariant* across a set of environments. Arjovsky et al. (2020) develop the invariant risk minimization (IRM) framework where environments are known, while Creager et al. (2021) propose environment inference for invariant learning (EIIL), to recover the invariant predictor, when the environments are not known. Predict

then interpolate (PI) (Bao et al., 2021) and BLOOD (Bae et al., 2022) use environment-specific ERMs to infer groups based on the correctness of predictions. We compare our approach with all these methods across several datasets to showcase its effectiveness in improving OOD generalization.

**Subgroup Robustness.** In many applications, models should do well not just on average but also on subgroups within the data. Several methods (Sagawa* et al., 2020; Nam et al., 2020; Kirichenko et al., 2022; Sohoni et al., 2022; Qiu et al., 2023; Zhang et al., 2022b) have been developed to improve the worst-group performance of a model. One widely used approach is to optimize the worst-case risk over a set of subgroups in the data (Duchi et al., 2019; Sagawa* et al., 2020; Setlur et al., 2023). CVaRDRO (Duchi et al., 2019) optimizes over all subgroups in the data, which is somewhat pessimistic, whereas GDRO (Sagawa* et al., 2020) does this over a set of predefined groups. Deng et al. (2023) begins with a group-balanced subset of training data and progressively expands it to encourage the learning of the task-relevant features. However, group knowledge may not always be available, various methods try to identify or infer groups and reweight minority groups in some way, when group labels are not available (Nam et al., 2020; Liu et al., 2021a; Sohoni et al., 2020) or partially available (Sohoni et al., 2022). Just train twice (JTT) (Liu et al., 2021a) uses ERM to identify the groups based on the correctness of predictions. Learning from failure (LfF) (Nam et al., 2020) simultaneously trains two NNs, encouraging one model to make biased predictions, and reweighting the samples it finds harder to learn (larger loss) to train the other model. We compare our approach with these methods on some benchmark datasets for this task from Table 1.

## 2 Spurious Features are Simple and Predictive

In this section, we conduct an experiment to validate our hypothesis that surrogate features are generally 'simpler' than invariant features (see Table 1 for examples). First, we define *simple models/features* for a task as follows:

**Definition 1** (Simple models and features)**.** *Consider a task on which benchmark models that can attain near-zero training loss have a certain complexity (in terms of the number of parameters, layers, etc.). We consider models that have significantly lower complexity than benchmark models and cannot attain small loss on the training data as simple models. Similarly, features that can be effectively learned using simple models are considered as simple features.*

Def. 1 proposes a metric of simplicity that is task-dependent since it depends on the complexity of the models which get high accuracy on the task. We choose to not define quantitative measures in Def. 1 since those would be problem-dependent. As an example, for colored-MNIST (CMNIST), NNs with non-linearities are necessary to achieve high accuracy, and for Waterbirds, deeper networks such as ResNet-50 (He et al., 2016) achieve the best results. Therefore, a linear model and a shallow CNN can be considered as simple models for these two datasets, respectively. We discuss simple model selection in more detail in Section 3.

Importantly, we observe that these simple models can still achieve good performance when the task is to distinguish between surrogate features even though they are not as accurate in predicting the invariant feature. Specifically, for CMNIST, we compare the performance of a linear model on color classification and digit classification on clean MNIST data. Similarly, for Waterbirds, we consider a shallow CNN (specifically, the `2DConvNet1` architecture in Appendix Fig. 10) and compare its performance on background classification (using images from the Places dataset (Zhou et al., 2017)) and bird type classification (using segmented images of birds from the CUB dataset (Wah et al., 2011)). The results in Table 2 corroborate the hypothesis that surrogate features for these datasets are simple features, as they can be predicted much more accurately by simpler models than invariant features. Unsurprisingly, the respective benchmark models can capture the task-relevant features much more accurately as they are more complex than the simple models used in each case.

Motivated by these observations, we define *spurious features* as follows. Operationally, our assumption on spurious features has the advantage that it does not require knowledge of some underlying causal graph or data from multiple environments to determine if a feature is spurious.

**Assumption 1** (Spurious features)**.** Spurious features *are simple features that are still reasonably correlated with the target label.*

| Dataset | Simple Model | | | | Benchmark Model | |
| | Predict surrogate feature | | Predict invariant feature | | Predict invariant feature | |
| | Train | Test | Train | Test | Train | Test |
|---|---|---|---|---|---|---|
| CMNIST | 100 | 100 | $86.2 \pm 0.2$ | $86.6 \pm 0.3$ | $97.3 \pm 0.1$ | $96.7 \pm 0.1$ |
| Waterbirds | $79.6 \pm 0.6$ | $78.4 \pm 0.6$ | $60.5 \pm 2.5$ | $60.4 \pm 2.4$ | $98.8 \pm 1.2$ | $96.2 \pm 1.1$ |

Table 2: Comparison between performance for predicting the simple feature and the complex feature.

| Dataset | Train | Test |
|---|---|---|
| CMNIST | $84.9 \pm 0.2$ | $10.7 \pm 0.3$ |
| Waterbirds | $93.3 \pm 0.5$ | $54.9 \pm 1.1$ |

Table 3: Train and test performance of the simple model on the target task.

In the presence of such features, simplicity bias leads the model to prefer such features over invariant features that are more complex. We also verify that when simple models are trained on the target tasks on these datasets, they tend to rely on these spurious features to make accurate predictions. Specifically, we consider the digit classification task using CMNIST data, where the correlation between the color and the label in the test set is 10%, and bird type classification using Waterbirds data, where the test set consists of balanced groups (50% correlation). Table 3 shows that the test accuracy is close to the correlation between the spurious feature-based group label and the target label. This indicates that simple models trained on the target task utilize spurious features to make predictions.

**What if Assumption 1 is not true?** We briefly note here that spurious features as defined by us may not always be irrelevant for the task, and mitigating simplicity bias may not always be desirable (further discussed with other limitations in Appendix D). Indeed, through similar experiments, we found that for the CelebA dataset, the spurious and task-relevant features are not significantly different in terms of complexity (details in Section 5.5). As a result, our approach does not lead to much improvement in the worst-group accuracy on this dataset (Section 5.3). However, it is impossible for any learning rule to generalize across all types of distribution shifts (see Appendix D for more discussion); one can only aim to generalize under specific types of distribution shifts by employing the appropriate inductive bias. Reducing reliance on simple, predictive features represents one such assumption, which seems reasonable based on the results presented above and proves effective in several cases, as shown in Section 5.

## 3 CMID: Learning in the Presence of Spurious Features

In this section, we outline our approach to mitigate simplicity bias. Our approach leverages the fact that simple models can capture surrogate features much better than invariant features (Table 2), and rely on the spurious features to make predictions, even when trained on the target task (Table 3). Thus, by ensuring that the final model has low conditional mutual information with respect to such a simple model, we encourage it to utilize a more diverse set of features.

Let $Z = (X, Y)$ denote an input-label pair, where $X \in \mathcal{X}, Y \in \{0, 1\}$, sampled from some distribution $\mathcal{D}$, $D$ denote a dataset with $n$ samples, $M(\theta) : \mathcal{X} \to [0, 1]$ denote a model, parameterized by $\theta$ (shorthand $M$). Subscripts $(\cdot)_s$ and $(\cdot)_c$ denote simple and complex, respectively. Let $\mathcal{M}$ denote the class of all models, $\ell_M(Z) : \mathcal{X} \times \{0, 1\} \to \mathbb{R}$ denote a loss function. $I(\cdot; \cdot)$ measures Shannon mutual information between two random variables. With slight abuse of notation, let $M$ and $Y$ also denote the (binary) random variables associated with the predictions of model $M$ on datapoints $X_i$'s and the labels $Y_i$'s, across all $i \in [n]$, respectively. The conditional mutual information (CMI) between the outputs of two models given the label is denoted by $I(M_1; M_2|Y)$. The empirical risk minimizer for class $\mathcal{M}$ is denoted as $M^* := \text{ERM}(\mathcal{M}) = \underset{M \in \mathcal{M}}{\arg \min} \frac{1}{n} \sum_{i \in [n]} \ell_M(Z_i)$.

We consider the class of simple models $\mathcal{M}_s \subset \mathcal{M}$ and complex models $\mathcal{M}_c = \mathcal{M} \setminus \mathcal{M}_s$. We now describe our proposed two-stage approach CMI Debiasing (CMID):

- First, learn a simple model $M_s$, which minimizes risk on the training data: $M_s^* = \text{ERM}(\mathcal{M}_s)$.

- Next, learn a complex model $M_c$ by regularizing its CMI with $M_s$:

$$M_c = \underset{M \in \mathcal{M}}{\arg \min} \frac{1}{n} \sum_{i \in [n]} \ell_M(Z) + \lambda \hat{I}_D(M; M_s^* | Y),$$

where $\hat{I}_D(M; M_s^*|Y)$ denotes the estimated CMI over $D$, and $\lambda$ is the regularization parameter.

A few remarks are in order about the choice of our regularizer and the methodology to select the simple model class. We penalize the CMI instead of MI. This is because both $M_s^*$ and $M$ are expected to have information about $Y$ (for e.g. in the Waterbirds dataset both bird type and background are correlated with the label). Hence they will not be independent of each other, but they are closer to being independent when conditioned on the label. We note that we use CMI to measure dependence, instead of other measures such as enforcing orthogonality of the predictions. This is for the simple reason that MI measures all—potentially non-linear—dependencies between the random variables.

We note that while several models can be considered simple based on Def. 1, the choice of the simple model class $\mathcal{M}_s$ used for our approach is task dependent. Intuitively, we want to use the simplest model that we expect to do reasonably well on the given task. Models that are too simple may not be able to capture surrogate features effectively, whereas models that are very complex may rely on task-relevant features, even though such reliance may be weak due to simplicity bias. As an example, Fig. 2 considers the problem of subgroup robustness on the Waterbirds dataset (where typically the hardest groups are images of waterbirds on land backgrounds and vice versa) and shows a comparison of the average and worst-group accuracy

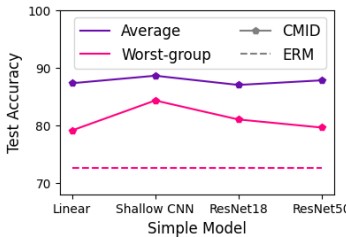

Figure 2: Performance of CMID using different simple model architectures for the Waterbirds dataset.

of the model learned by CMID using various simple model architectures (see Section 5.3 for more details about the task). We consider four models: a linear model, the shallow CNN (`2DConvNet1`) used in Table 2, a ResNet-18 pre-trained on ImageNet, and the ResNet-50 pre-trained on ImageNet, which is also the architecture of the benchmark model. We observe that using a shallow CNN as the simple model is most effective for CMID. This is because a linear model is too simple to capture the surrogate feature, i.e. the background in this case, whereas deep models like ResNet are complex enough to learn both types of features while relying more strongly on the surrogate feature. As a result, regularizing the CMI with respect to such models may not be as effective in reducing reliance on spurious features. However, we note that doing so still leads to significantly better worst-group accuracy values compared to ERM.

Although the choice of $\mathcal{M}_s$ may impact the performance of CMID, we use a fairly simple selection rule throughout our experiments, which works well. In general, for datasets where the final model is a shallow NN, we used a linear model as the simple model. For image datasets where the final model is a ResNet or DenseNet-based model, we consider shallow 2D CNNs as simple models. For language datasets where BERT-based models are the final models, we consider shallow MLPs or 1D CNNs as simple models. Details about the model architectures can be found in Appendix G.3.2.

We also note that the CMI is not differentiable in its original form since $Y$ is discrete. Therefore, we employ an estimator that uses the predicted probabilities instead of the thresholded predictions to make the regularization differentiable and hence suitable for gradient-based methods. We include details for this estimator and also an empirical comparison with the original CMI in Appendix A.

## 4 Theoretical Results

In this section, we analyze the effect of CMI regularization and show it leads to reduced dependence on spurious features in a Gaussian mixture model. We also obtain a simple OOD generalization guarantee for CMID in a causal learning framework.

### 4.1 Effect of CMI Regularization

We consider data generated from the following Gaussian mixture model (see the left-most panel in Fig. 3 for an example). We note that Sagawa et al. (2020); Rosenfeld et al. (2021) consider a similar Gaussian mixture-based data model for invariant and spurious features.

**Assumption 2.** *Let the label* $y \sim \mathcal{R}(0.5)$, *where* $\mathcal{R}(p)$ *is a* $\{\pm 1\}$ *random variable which is 1 with probability* $p$. *Consider* $d$ *invariant features* $X_1, \ldots, X_d$ *and a spurious feature* $X_{d+1}$ *such that* $X_i \perp X_j \mid y$ *for all* $i, j \in [d+1], i \neq j$, *with distributions,*

$$X_i \sim \mathcal{N}(y\mu_1, \sigma_1^2) \ \forall i \in [d], \ and \ X_{d+1} \sim \mathcal{N}(a\mu_2, \sigma_2^2),$$

*where* $a \sim y\mathcal{R}(\eta)$ *is a spurious attribute, with an unstable correlation with* $y$, *and* $\mu_1, \mu_2 > 0, \eta > 0.5$. *Let* $\mu_2' := (2\eta - 1)\mu_2$, $\sigma_2'^2 := \sigma_2^2 + \mu_2^2 - \mu_2'^2$.

We consider linear models to predict $y$ from the features $X_1, \ldots, X_{d+1}$. Let $\mathcal{M} = \{\boldsymbol{w} = (w_1, \ldots, w_{d+1}) : \boldsymbol{w} \in \mathbb{R}^{d+1}\}$ be all possible linear models and $\mathcal{M}_s = \{\boldsymbol{w} = (w_1, \ldots, w_{d+1}) : \boldsymbol{w} \in \mathbb{R}^{d+1}, \|\boldsymbol{w}\|_0 = 1\}$ be a simpler model class which only uses one of the $d+1$ features. We consider the mean squared error (MSE) loss, and the ERM solution is given by:

$$\text{ERM}(\mathcal{M}) = \underset{\boldsymbol{w} \in \mathcal{M}}{\arg\min} \ \mathbb{E} \left( \sum_{i=1}^{d+1} w_i X_i - y \right)^2.$$

**Proposition 1.** *For all* $i \in [d]$, *ERM($\mathcal{M}$) satisfies* $\frac{w_i}{w_{d+1}} = \frac{\mu_1}{\mu_2'} \frac{\sigma_2'^2}{\sigma_1^2}$. *When* $\frac{\mu_1}{\mu_2'} \frac{\sigma_2'^2}{\sigma_1^2} < 1$, *ERM($\mathcal{M}_s$)* $= \left[ 0, \ldots, 0, \frac{\mu_2'}{\sigma_2^2} \right]$ *(upto scaling).*

We now consider the effect of CMI regularization. We consider the case when $\frac{\mu_1}{\mu_2'} \frac{\sigma_2'^2}{\sigma_1^2} < 1$, so the spurious feature has a higher signal-to-noise ratio than any individual invariant feature. Then, as per Proposition 1 in the first step we learn a simple model $w_{d+1}^* X_{d+1}$ which uses only $X_{d+1}$. We now consider the ERM problem but with a constraint on the CMI:

$$\text{ERM}_\mathcal{C}(\mathcal{M}) = \underset{\boldsymbol{w} \in \mathcal{M}}{\arg\min} \ \mathbb{E} \left( \sum_{i=1}^{d+1} w_i X_i - y \right)^2$$

$$\text{s.t.} \ I \left( \sum_{i=1}^{d+1} w_i X_i; w_{d+1}^* X_{d+1} | y \right) \leq \nu. \tag{1}$$

We show the following guarantee on the learned model.

**Theorem 1.** *Let data be generated as per Assumption 2. For* $\nu = 0.5 \log(1 + c^2)$ *for some* $c$:

1. *When* $\frac{\mu_1}{\mu_2'} \frac{\sigma_2'^2}{\sigma_1 \sigma_2} > \frac{1}{c\sqrt{d}}$, *the solution to (1) is the same as ERM($\mathcal{M}$), so for all* $i \in [d]$, $\frac{w_i}{w_{d+1}} = \frac{\mu_1}{\mu_2'} \frac{\sigma_2'^2}{\sigma_1^2}$.

2. *Otherwise, for all* $i \in [d]$, $w_i$ *is upweighted and the solution to (1) satisfies* $\frac{|w_i|}{|w_{d+1}|} = \frac{1}{c\sqrt{d}} \frac{\sigma_2}{\sigma_1}$.

Theorem 1 suggests that for an appropriately small $c$, regularizing CMI with the simple model leads to a predictor which mainly uses the invariant features. This is supported by experimental results on data drawn according to Assumption 2 with $d = 1$, shown in Fig. 3. We note that a lower value of $c$ promotes conditional independence with $X_{d+1}$ and upweighs $w_i$ more strongly. When $c \to 0$, $w_{d+1} \to 0$. We visualize this in Fig. 9 in the Appendix for $d = 1$. In Appendix E.1, we show a similar theoretical result when multiple spurious features are present.

## 4.2 OOD Generalization in a Causal Learning Framework

Following the setting in Arjovsky et al. (2020); Liu et al. (2021b), we consider a dataset $D = \{D^e\}_{e \in \mathcal{E}_{tr}}$, which is composed of data $D^e \sim \mathcal{D}_e^{n_e}$ gathered from different training environments $e \in \mathcal{E}_{tr}$, where $e$ denotes an environment label, $n_e$ represents the number of samples in $e$. $\mathcal{E}_{tr}$ denotes the set of training environments.

The problem of finding a predictor with good OOD generalization performance can be formalized as:

$$\underset{M \in \mathcal{M}}{\arg\min} \ \underset{e \in \mathcal{E}}{\max} \ \mathbb{E}_D[\ell_M(Z)|e],$$

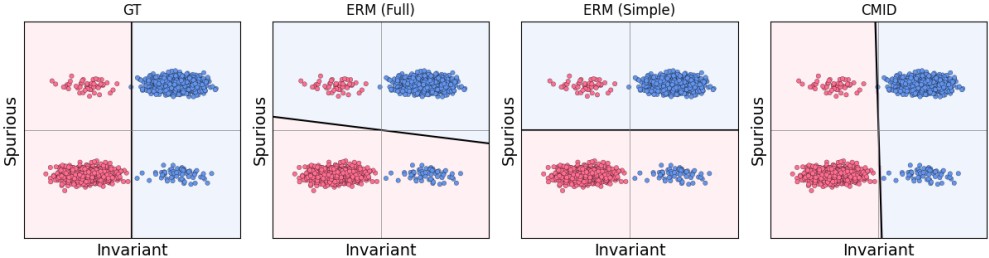

Figure 3: Results on synthetic Gaussian data generated as per Assumption 2, with 2000 samples, $d = 1, \mu_1 = \mu_2 = 5$, $\sigma_1 = 1.5, \sigma_2 = 0.5$. Left to right: Ground truth (GT) predictor, ERM with $\mathcal{M}$ as the class of linear models, ERM over the class of threshold functions, ERM with CMI constraint, $c = 0.01$.

*i.e.*, optimizing over the worst-case risk on all environments in set $\mathcal{E}$. Usually, $\mathcal{E} \supset \mathcal{E}_{tr}$, and hence, the data and label distribution can differ significantly for $e \in \mathcal{E}_{tr}$ and $e \in \mathcal{E} \setminus \mathcal{E}_{tr}$. This makes the OOD generalization problem hard to solve.

The invariant learning literature assumes the existence of invariant and variant features. In this section, we assume that the model of interest, say $M(X)$ is composed of a featurizer $\Phi$ and a classifier $\omega$ on top of it, *i.e.* $M(X) = \omega \circ \Phi(X)$. For simplicity, we omit the argument $X$ and assume that learning a featurization includes learning the corresponding classifier, so we can write $M = \Phi$. Let $E$ be a random variable sampled from a distribution on $\mathcal{E}$.

**Definition 2** (Invariant and Variant Predictors)**.** *A feature map $\Phi$ is called* invariant *and is denoted by $\Phi$ if $Y \perp E | \Phi$, whereas it is called* variant *and is denoted by $\Psi$ if $Y \not\perp E | \Psi$.*

Several works (Arjovsky et al., 2020; Liu et al., 2021b; Creager et al., 2021) attempt to recover the invariant feature map by proposing different ways to find the maximally invariant predictor (Chang et al., 2020), defined as:

**Definition 3** (Invariance Set and Maximal Invariant Predictor)**.** *The invariance set $\mathcal{I}$ with respect to environment set $\mathcal{E}$ and hypothesis class $\mathcal{M}$ is defined as:*

$$\mathcal{I}_{\mathcal{E}}(\mathcal{M}) = \{\Phi : Y \perp E | \Phi\} = \{\Phi : H[Y|\Phi] = H[Y|\Phi, E]\}.$$

*The corresponding maximal invariant predictor (MIP) of $\mathcal{I}_{\mathcal{E}}(\mathcal{M})$ is $\Phi^* = \underset{\Phi \in \mathcal{I}_{\mathcal{E}}(\mathcal{M})}{\arg\max} I(Y; \Phi)$.*

The MIP is an invariant predictor that captures the most information about $Y$. Invariant predictors guarantee OOD generalization, making MIP the optimal invariant predictor (Theorem 2.1 in Liu et al. (2021b)).

As discussed in Section 1, most current work assumes that the environment labels $e$ for the datapoints are known. However, environment labels typically are not provided in real-world scenarios. In this work, we don't assume access to environment labels and instead, we rely on another aspect of these features: are they simple or complex? We formalize this below:

**Assumption 3** (Simple and Complex Predictors)**.** *The invariant feature comprises of complex features, i.e. $\Phi^* = [\Phi_c]$, where $\Phi_c \in \mathcal{M}_c$. The variant feature comprises of simple and complex features, i.e., $\Psi^* = [\Psi_c, \Psi_s]$, where $\Psi_c \in \mathcal{M}_c$, $\Psi_s \in \mathcal{M}_s$ and $I(Y; \Psi_s) > 0$.*

We consider the underlying causal model in Rosenfeld et al. (2021) which makes the following assumption.

**Assumption 4** (Underlying Causal Model)**.** *Given the model in Fig. 4, $I(\Phi, \Psi|Y) = 0$ and $I(\Psi_s, \Psi_c|Y) > I(\Psi_s, \Psi_c|Y, E)$.*

The following simple result shows that our method finds the maximal invariant predictor, and thus generalizes OOD.

**Proposition 2.** *Let $ERM(\mathcal{M}_s) = M_s^*$. Under Assumptions 3 and 4, the solution to the problem:*

$$\arg\min_{M \in \mathcal{M}} \mathbb{E}\, \ell_M(Z) \;\; s.t. \;\; I(M; M_s^*|Y) = 0 \qquad (2)$$

*is $M = \Phi^*$, the maximal invariant predictor.*

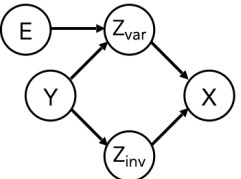

Figure 4: Our causal model. Latent variables $Z_{\text{inv}}$ and $Z_{\text{var}}$ correspond to invariant and variant features $\Phi^*$ and $\Psi^*$ respectively.

Here, we note that Assumption 3 is key to our result. Although this assumption may not always hold directly in practice, it characterizes the condition under which our approach can recover the invariant predictor even though it does not have an explicit causal/invariant learning motivation or access to environment labels.

## 5 Experiments

We show that CMID reduces simplicity bias, and yields improvements across various robustness, OOD generalization and fairness metrics. We first show that CMID mitigates simplicity bias in Slab and ImageNet-9 datasets. We then evaluate CMID on various OOD generalization tasks. These include data with multiple spurious features, real-world medical data, and a real-world fairness application. Finally, we test CMID on some benchmark datasets for subgroup robustness and a fairness application. [2]. We note that past approaches usually target one or two of these problem settings. Thus, in each section, we generally choose the most task-relevant methods to compare with, as established in prior work on that task. For consistency, in addition to ERM, we compare with JTT on most of the tasks, since it is the most similar method to CMID in terms of its requirements. Throughout, we observe that CMID improves considerably on ERM and compares favorably with JTT and domain-specific approaches on most data.

### 5.1 Mitigating Simplicity Bias

**Slab Data.** Slab data was proposed in Shah et al. (2020) to model simplicity bias. Each feature is composed of $k$ data blocks or slabs. We consider two configurations of the slab data, namely 3-Slab and 5-Slab, as shown in Fig. 5. In both cases, the first feature is linearly separable. The second feature has 3 slabs in the 3-Slab data and 5 slabs in the 5-Slab data. The first feature is simple since it is linearly separable, while the features with more slabs involve a piece-wise linear model and are thus complex. The linear model is perfectly predictive, but the predictor using both types of features attains a much better margin, and generalizes better under fixed $\ell_1$-norm perturbations to the features. Fig. 5 shows the decision boundary using

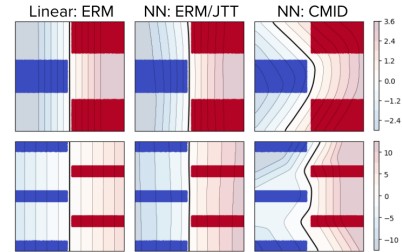

Figure 5: Results on the slab dataset, when training a linear model and a 1-hidden-layer NN using ERM, JTT, and CMID.

ERM/JTT and the proposed approach. We see that CMID encourages the model to use both features and attain better margin. We note that since both the features are fully predictive in this setting, approaches like JTT are ineffective as they rely on incorrect predictions of one model to train another model by upweighting such samples.

**Texture vs Shape Bias on ImageNet-9.** Geirhos et al. (2022) showed that CNNs trained on ImageNet tend to make predictions based on image texture rather than image shape. To quantify this phenomenon, the authors designed the GST dataset, which contains synthetic images with conflicting shapes and textures (e.g., an image of a cat modified with elephant skin texture as a conflicting cue). The *shape bias* of a model on the GST dataset is defined as the number of samples for which the model correctly identifies shape compared to the total number of samples for which the model correctly identifies either shape or texture. A shape bias of 100 indicates that the model always uses shape when shape conflicts with texture, whereas 0 indicates that the model always uses texture instead of shape.

---

[2]Our code is available at `https://github.com/estija/CMID`.

| Method | ERM | JTT | CMID |
|---|---|---|---|
| Shape Bias | $31.0_{\pm1.3}$ | $39.9_{\pm0.2}$ | $\mathbf{40.2}_{\pm3.6}$ |
| ImageNet-A Acc | $32.6_{\pm0.6}$ | $30.7_{\pm2.6}$ | $\mathbf{35.5}_{\pm2.6}$ |

Table 4: Shape bias and OOD accuracy on ImageNet-9 dataset.

| Method | Group labels | Bias: Color | | | Bias: Color & Patch | | |
|---|---|---|---|---|---|---|---|
| | | Test (i.i.d) $p_e = 0.1$ | Test (OOD) $p_e = 0.9$ | $\delta_{gap}$ | Test (i.i.d) $p_e = 0.1$ | Test (OOD) $p_e = 0.9$ | $\delta_{gap}$ |
| ERM | No | $88.6_{\pm0.3}$ | $16.4_{\pm0.8}$ | $-72.2$ | $93.7_{\pm0.3}$ | $14.0_{\pm0.5}$ | $-79.7$ |
| EIIL (Creager et al., 2021) | No | $71.7_{\pm1.6}$ | $62.8_{\pm5.0}$ | $-8.9$ | $65.3_{\pm8.4}$ | $53.0_{\pm5.6}$ | $-12.3$ |
| JTT (Liu et al., 2021a) | No | $72.2_{\pm1.1}$ | $\underline{64.6}_{\pm0.56}$ | $-7.6$ | $64.0_{\pm2.7}$ | $\underline{56.2}_{\pm2.7}$ | $-7.8$ |
| CMID | No | $69.2_{\pm0.9}$ | $\mathbf{68.9}_{\pm0.9}$ | $-0.3$ | $60.3_{\pm2.7}$ | $\mathbf{59.4}_{\pm1.0}$ | $-0.9$ |
| IRM (Arjovsky et al., 2020) | Yes | $71.4_{\pm0.9}$ | $66.9_{\pm2.5}$ | $-4.5$ | $93.5_{\pm0.2}$ | $13.4_{\pm0.3}$ | $-80.1$ |
| GDRO (Sagawa* et al., 2020) | Yes | $89.2_{\pm0.9}$ | $13.6_{\pm3.8}$ | $-75.6$ | $92.3_{\pm0.3}$ | $14.1_{\pm0.8}$ | $-78.2$ |
| PI (Bao et al., 2021) | Yes | $70.3_{\pm0.3}$ | $70.2_{\pm0.9}$ | $-0.1$ | $85.4_{\pm0.9}$ | $15.3_{\pm2.7}$ | $-70.1$ |
| BLOOD (Bae et al., 2022) | Yes | $70.5_{\pm1.1}$ | $70.7_{\pm1.4}$ | $0.2$ | $68.3_{\pm2.3}$ | $62.3_{\pm3.3}$ | $-6.0$ |
| Optimal | - | $75$ | $75$ | $0$ | $75$ | $75$ | $0$ |

Table 5: Comparison of average test accuracies on i.i.d and OOD data and their difference ($\delta_{gap}$) on Colored MNIST and Color+Patch MNIST datasets. **Bold** and underlined numbers indicate the best and second-best OOD performance among the methods that don't use group labels.

We consider ImageNet-9, a subset of ImageNet organized into nine classes by Xiao et al. (2020) (details in Appendix G.1.2) and train a ResNet18 model. We test OOD model performance on ImageNet-A (Hendrycks et al., 2021), a set of natural adversarial examples on which existing ImageNet classifiers perform poorly. Table 4 shows that CMID mitigates texture bias, resulting in improved performance on adversarial examples.

## 5.2 Better OOD Generalization

**Synthetic: CMNIST and Color+Patch MNIST.** We present results on two variants of the MNIST dataset (Deng, 2012), which contains images of handwritten digits, using a binary digit classification task ($< 5$ or not). The colored-MNIST data was proposed in Arjovsky et al. (2020), where color (red/green) is injected as a spurious feature, with unstable correlation $1 - p_e$ with the label across environments. The train data has two environments with $p_e = 0.1, 0.2$ while the test data has $p_e = 0.9$, to test OOD performance. Further, it contains 25% label noise to reduce the predictive power of the task-relevant feature: digit shape. We also consider the color+patch MNIST data proposed in Bae et al. (2022), where an additional spurious feature is injected into the data, in the form of a $3 \times 3$ patch. The position of the patch (top left/bottom right) is correlated with the label, with the same $p_e$, but independent of the color.

Following Bae et al. (2022), we evaluate on i.i.d test data with $p_e = 0.1$ and OOD data with $p_e = 0.9$ for both the cases (details in Appendix G.2.1). Table 5 shows that CMID gets competitive OOD performance with methods that require group knowledge, and has the lowest gap $\delta_{gap}$ between test performance on i.i.d and OOD samples, even in the presence of multiple spurious features.

**Medical: Camelyon17-WILDS.** Camelyon17-WILDS is a real-world medical image dataset of data collected from five hospitals (Bándi et al., 2019; Koh et al., 2021). Three hospitals comprise the training set, one is the validation set and the third is the OOD test set. Images from different hospitals vary visually. The task is to predict whether or not the image contains tumor tissue, and the dataset is a well-known OOD generalization benchmark (Bae et al., 2022; Koh et al., 2021). Table 6 shows that CMID leads to higher

| Method | ERM | IRM | GDRO | PI | BLOOD | JTT | CMID |
|---|---|---|---|---|---|---|---|
| Train Acc | $97.3_{\pm0.1}$ | $97.1_{\pm0.1}$ | $96.5_{\pm1.4}$ | $93.2_{\pm0.2}$ | $93.0_{\pm1.8}$ | $88.4_{\pm1.3}$ | $94.0_{\pm2.0}$ |
| OOD Test Acc | $66.5_{\pm4.2}$ | $59.4_{\pm3.7}$ | $70.2_{\pm7.3}$ | $71.7_{\pm7.5}$ | $74.9_{\pm5.0}$ | $\mathbf{78.0}_{\pm6.3}$ | $\underline{77.9}_{\pm7.7}$ |

Table 6: Comparison of average train and OOD test accuracies on Camelyon17-WILDS dataset.

| Method | ERM | ARL | JTT | EIIL | CMID |
|---|---|---|---|---|---|
| Train Acc | $92.7_{\pm0.5}$ | $72.1_{\pm3.6}$ | $80.2_{\pm1.7}$ | $69.7_{\pm1.6}$ | $76.2_{\pm2.2}$ |
| OOD Test Acc | $31.1_{\pm4.4}$ | $61.3_{\pm1.7}$ | $71.8_{\pm5.3}$ | $\mathbf{78.8}_{\pm1.4}$ | $\mathbf{78.8}_{\pm0.7}$ |
| $\delta_{gap}$ | $-61.6$ | $-10.8$ | $-8.4$ | $9.1$ | $2.6$ |

Table 7: Comparison of average train and OOD test accuracies and their difference ($\delta_{gap}$) on Adult-Confounded dataset.

| Method | Group labels | Waterbirds | | CelebA | | MultiNLI | | CivilComments-WILDS | |
|---|---|---|---|---|---|---|---|---|---|
| | | Average | Worst-group | Average | Worst-group | Average | Worst-group | Average | Worst-group |
| ERM | No | 97.3 | 72.6 | 95.6 | 47.2 | 82.4 | 67.9 | 92.6 | 57.4 |
| CVaRDRO | No | 96.0 | 75.9 | 82.5 | 64.4 | 82.0 | 68.0 | 92.5 | 60.5 |
| LfF | No | 91.2 | 78.0 | 85.1 | 77.2 | 80.8 | 70.2 | 92.5 | 58.8 |
| JTT | No | 93.3 | **86.7** | 88.0 | **81.1** | 78.6 | **72.6** | 91.1 | 69.3 |
| CMID | No | 88.6 | 84.3 | 84.5 | 75.3 | 81.4 | 71.5 | 84.2 | **74.8** |
| GDRO | Yes | 93.5 | 91.4 | 92.9 | 88.9 | 81.4 | 77.7 | 88.9 | 69.9 |

Table 8: Average and worst-group test accuracies on benchmark datasets for subgroup robustness. **Bold** and underlined numbers indicate the best and second-best worst-group accuracy among the methods that don't use group labels.

average accuracies than existing group-based methods when evaluated on images from the test hospital. While JTT attains a comparable test accuracy, it exhibits a much larger gap between ID and OOD accuracy compared to CMID.

**Fairness: Adult-Confounded.**  The Adult-Confounded dataset is a semi-synthetic variant of the UCI Adult dataset (Newman et al., 1998; Leisch & Dimitriadou, 2021), developed by Creager et al. (2021). It consists of four sensitive subgroups based on binarized race and sex labels, and has confounded data where subgroup membership is predictive of the label on the training data but the correlation is reversed at test time. Therefore, it tests whether the classifier makes biased predictions based on subgroup membership.

Table 7 shows that compared to other methods CMID achieves superior OOD test performance with the least gap between train and test performance, indicating its low reliance on sensitive subgroup information.

## 5.3 Subgroup Robustness

We evaluate our approach on four benchmark classification tasks for robustness to spurious correlations, namely on Waterbirds (Sagawa* et al., 2020), CelebA (Liu et al., 2015), MultiNLI (Williams et al., 2018) and CivilComments-WILDS (Borkan et al., 2019) datasets (Table 1, details in Appendix G.3).

Although CMID does not require group knowledge for training, following Liu et al. (2021a), we use a validation set with group labels for model selection. Table 8 shows the average and worst-group accuracies for CMID and comparison with other methods (Duchi et al., 2019; Nam et al., 2020; Liu et al., 2021a) which do not use group information. GDRO (Sagawa* et al., 2020), which uses group information, acts as a benchmark. We see that on three of these datasets, CMID competes with state-of-the-art algorithms that improve subgroup robustness. Interestingly, CMID seems particularly effective on the two language-based datasets. We also note that CMID is not very effective on CelebA images. We believe that this is because both the spurious feature (gender) and the invariant feature (hair color) for CelebA are of similar complexity. In Section 5.5, we explore this further with a similar experiment as in Section 2 for CelebA.

## 5.4 Fairness Application: Bias in Occupation Prediction

The Bios dataset (De-Arteaga et al., 2019; Cheng et al., 2023) is a large-scale dataset of more than 300k biographies scrapped from the internet. The goal is to predict a person's occupation based on their bio. Based on this task, Cheng et al. (2023) formalizes a notion of social norm bias (SNoB). SNoB captures the extent to which predictions align with gender norms associated with specific occupations. In addition to gender-specific pronouns, these norms encompass other characteristics mentioned in the bios. They represent implicit expectations of how specific groups are expected to behave. Cheng et al. (2023) characterizes SNoB

| Method | ERM | Decoupled | CMID |
|--------|-----|-----------|------|
| Accuracy | 0.95 | 0.94 | **0.96** |
| $\rho(p_c, r_c)$ | 0.66 | 0.60 | **0.38** |

Table 9: Comparison of accuracy in occupation prediction and the correlation between gender and occupation prediction on Bios data.

| Predict invariant feature | | Predict surrogate feature | |
|---------|---------|---------|---------|
| Train | Test | Train | Test |
| $89.1 \pm 1.5$ | $84.3 \pm 0.6$ | $92.4 \pm 2.4$ | $88.3 \pm 1.6$ |

Table 10: Comparison between performance for predicting the simple feature and the complex feature on CelebA dataset.

as a form of algorithmic unfairness arising from the associations between an algorithm's predictions and individuals' adherence to inferred social norms. They also show that adherence to or deviations from social norms can result in harm in many contexts and that SNoB can persist even after the application of some fairness interventions. We note that SNoB is a distinct type of bias, and existing de-biasing methods have not been evaluated on this task.

To quantify SNoB, the authors utilize the Spearman rank correlation coefficient $\rho(p_c, r_c)$, where $p_c$ represents the fraction of bios associated with occupation $c$ that mention the pronoun 'she', and $r_c$ measures the correlation between occupation predictions and gender predictions. The authors employ separate one-vs-all classifiers for each occupation and obtain the occupation prediction for a given bio using these classifiers. For gender predictions, they train occupation-specific models to determine the gender-based group membership (female or not) based on a person's bio, and use the predictions from these models. A higher value of $\rho(p_c, r_c)$ represents a larger social norm bias, which indicates that in male-dominated occupations, the algorithm achieves higher accuracy on bios that align with inferred masculine norms, and vice-versa. Table 9 shows results on the Bios data. We compare with a group fairness approach, Decoupled (Dwork et al., 2018) that trains separate models for each gender, in order to mitigate gender bias. We see that CMID addresses SNoB bias better than ERM and Decoupled, achieving a lower $\rho(p_c, r_c)$ and improved accuracy.

### 5.5 Additional Experiment: Invariant and Spurious Features have Similar Complexity for CelebA

In our subgroup robustness experiment for CelebA (Table 8), we found that our method did not yield a significant improvement in worst-group accuracy. We investigate this further in this section. We show that for the CelebA dataset, the complexity of the invariant and surrogate features are actually quite similar. The experiment is similar to the experiments we did for CMNIST and Waterbirds in Section 2. We create a subset of the CelebA dataset by sampling an equal number of samples from all four subgroups. Table 10 presents a comparison of the results when predicting the invariant feature (hair color) and the surrogate feature (gender) using the simple model (`2DConvNet2`). We observe that the performance for both tasks is comparable, suggesting that the features exhibit similar complexity.

We contrast these results with those for CMNIST and Waterbirds in Table 2. For CMNIST and Waterbirds, there was a significant difference in the accuracy with which the simple model could predict the invariant and surrogate features. For CelebA, the difference is much smaller which suggests that spurious features are *not* simpler than invariant features for this dataset—explaining why our method is not as effective for it.

## 6 Conclusion

Our work presented a hypothesis about spurious features connected to simplicity bias: spurious features are simple features that are still predictive of the label, which proved to be valid across several datasets. We leveraged this hypothesis to propose a new framework (CMID) to mitigate simplicity bias and showed that it yields improvements over ERM and many other previous approaches across a number of OOD generalization, robustness and fairness benchmarks. It would be interesting to consider other natural definitions of spurious features (some of which may apply even when our definition is not suitable), and to build a comprehensive and principled framework for distributional robustness under suitable assumptions. On the algorithmic side, our work suggests that using simple models to audit and refine much larger models can lead to improved properties of the larger model along certain robustness and fairness axes. It would be interesting to explore the power of similar approaches for other desiderata and requirements that arise in trustworthy machine learning, and to understand its capabilities for fine-tuning large pre-trained models.

## Acknowledgements

This work was supported by NSF CAREER Award CCF-2239265 and an Amazon Research Award. The authors acknowledge use of the Amazon Elastic Compute Cloud and USC Advanced Research Computing's Discovery cluster. BV thanks Puneesh Deora for helpful discussions and feedback, and Deqing Fu for discussion on simplicity bias.

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

## Appendix

## A  Estimating CMI

In this section, we describe our technique to compute the CMI regularization for our approach, as mentioned in Section 3. We begin with the relevant notation.

Let $Z = (X, Y)$ denote an input-label pair, where $X \in \mathcal{X}, Y \in \{0, 1\}$, sampled from some distribution $\mathcal{D}$, $D$ denote a dataset with $n$ samples, $M(\theta) : \mathcal{X} \to [0, 1]$ denote a model, parameterized by $\theta$ (shorthand $M$). The predictions of the model are given by $\mathbb{1}[M(X) > 0.5]$. Subscripts $(\cdot)_s$ and $(\cdot)_c$ denote simple and complex, respectively. Let $\mathcal{M}$ denote the class of all models $M$, $\ell_M(Z) : \mathcal{X} \times \{0, 1\} \to \mathbb{R}$ denote a loss function. Let $H(\cdot)$ denote the Shannon entropy of a random variable. $I(\cdot; \cdot)$ measures Shannon mutual information between two random variables. With slight abuse of notation, let $M$ and $Y$ also denote the (binary) random variables associated with the predictions of model $M$ on datapoints $X_i$'s and the labels $Y_i$'s, across all $i \in [n]$, respectively. Let $\sigma(x, T) = \dfrac{1}{1 + e^{-Tx}}$ denote the sigmoid function, with temperature parameter $T$.

To estimate CMI, first consider the conditional (joint) distributions over $n$ samples:

$$p(M\!=\!m|Y\!=\!y) = \frac{\Sigma_{i\in[n]}\mathbb{1}\left[Y_i\!=\!y\right]\zeta(M(X_i),m)}{\Sigma_{i\in[n]}\mathbb{1}\left[Y_i\!=\!y\right]},$$

$$p(M\!=\!m, M_s\!=\!m'|Y\!=\!y) = \frac{\Sigma_{i\in[n]}\mathbb{1}\left[Y_i\!=\!y\right]\zeta(M(X_i),m)\,\zeta(M_s(X_i),m')}{\Sigma_{i\in[n]}\mathbb{1}\left[Y_i\!=\!y\right]},$$

where $m, m', y \in \{0, 1\}$, $\zeta(M(X_i), 1)\!=\!\mathbb{1}[M(X_i)\!>\!0.5]$, and $\zeta(M(X_i), 0)\!=\!1-\mathbb{1}[M(X_i)\!>\!0.5]$.

Note that the CMI computed using these would not be differentiable as these densities are computed by thresholding the outputs of the model. Since we want to add a CMI penalty as a regularizer to the ERM objective and optimize the proposed objective using standard gradient-based methods, we need a differentiable version of CMI. Thus, for practical purposes, we use an approximation of the indicator function $\mathbb{1}[x > 0.5]$, given by $\sigma(x - 0.5, T)$, where $T$ determines the degree of smoothness or sharpness in the approximation.

This can be easily generalized for multi-class classification with $C$ classes. In that case, $m, m', y \in \{0, \cdots, C-1\}$, $M(X_i)$ is a $C$-dimensional vector with the $m^{\text{th}}$ entry indicating the probability of predicting class $m$, and $\zeta(M(X_i), m)\!=\!\mathbb{1}[\arg\max_{j\in[C]} M_j(X_i) = m]$. To make this differentiable, we use the softmax function with temperature parameter $T$ to approximate the indicator function.

Using these densities, the estimated CMI is:

$$\hat{I}_n(M, M_s|Y)\!=\!\sum_y p(Y\!=\!y_i)\sum_{m,m'} p(M=m, M_s=m'|Y=y)\log\left[\frac{p(M\!=\!m, M_s\!=\!m'|Y\!=\!y)}{p(M\!=\!m|Y=y)p(M_s\!=\!m'|Y=y)}\right]. \tag{3}$$

This estimate is differentiable, making it compatible with gradient-based methods. Therefore, we utilize it as a regularizer for the proposed approach.

## A.1 Evaluating the Estimated CMI

In this section, we evaluate the reliability and scalability of the CMI estimate in (3) compared to the original CMI, which is computed with discretized model outputs. We consider CMNIST data with the hyperparameter values as mentioned in Section G.2.1 for the results in this section. Table 11 compares the times (in milliseconds) to compute the original CMI and the estimated CMI, using batch size 64, for 10 classes and 200 classes. We see

| Method | 10 classes | 200 classes |
|---|---|---|
| $I(M, M_s|Y)$ | $14.1 \pm 3.3$ | $23.3 \pm 2.7$ |
| $\hat{I}_n(M, M_s|Y)$ | $20.17 \pm 4.2$ | $24.3 \pm 3.3$ |

Table 11: Comparison of computation times (in milliseconds) for the original CMI and the estimated CMI in (3).

that the computation time for the estimated CMI does not increase significantly as the number of classes increases.

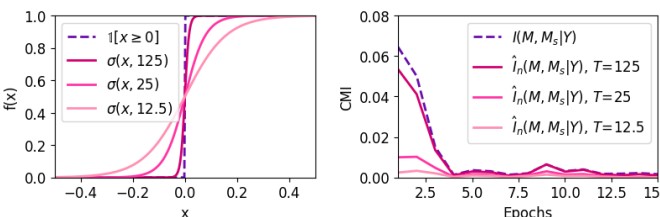 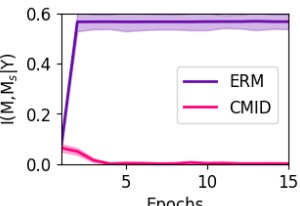

Figure 6: (Left) Sigmoid functions for different values of the temperature parameter $T$. (Right) Comparison of the original CMI with the estimated CMI using different values of $T$.

Figure 7: Comparison of the original CMI for models trained with ERM and CMID as a function of training time.

Fig. 6 compares the estimated CMI for different values of the temperature parameter $T$ with the original CMI. The values shown in the figure are computed with batch size 1000, while for training, we use the estimated

CMI with $T = 12.5$ and batch size 64. We see that as $T$ increases, the sigmoid approximates the indicator function more closely and the estimated CMI tends closer to the original CMI. Fig. 7 compares the CMI computed with discretized outputs using batch size 1000, when the model is trained with ERM or CMID. We see that regularizing the estimated CMI causes the original CMI to decrease over time, whereas training with ERM leads to an increase in the CMI. This shows that the solution learned by ERM makes predictions aligned with those of the simple model, whereas our approach CMID learns a model that attains low CMI with the simple model.

## A.2 Effect of the Simple Model Architecture

In Fig. 2 (Section 3), we compared the effect of using various simple model architectures on the performance of the model learned by CMID on the Waterbirds dataset. In this section, we conduct a similar experiment to analyze the effect of using various simple model architectures on the CMNIST dataset. Fig. 8 compares the performance of CMID when using a linear model and an MLP as the simple model. We observe that using a linear model is more effective in this case since the benchmark model for this task is an MLP. As a result, the MLP may capture both features, and regularizing the CMI with respect to this model may not be very effective in reducing the reliance on just the spurious features.

Figure 8: Performance of CMID using different simple model architectures for CMNIST dataset.

|  | Waterbirds | | | | CMNIST | |
|---|---|---|---|---|---|---|
|  | Linear | Shallow CNN | ResNet18 | ResNet50 | Linear | MLP |
| LR | $5 \times 10^{-5}$ | $10^{-5}$ | $5 \times 10^{-6}$ | $2 \times 10^{-5}$ | 0.01 | 0.01 |

Table 12: Training details for the simple models.

We note that in these experiments (for both Waterbirds and CMNIST datasets), we only tune the learning rate (LR) and weight decay ($\lambda_2$) for training the simple model (as listed in Table 12). The rest of the hyperparameter values for training the final model are kept consistent across each dataset (as listed in Table 17 for CMNIST and Table 19 for Waterbirds).

# B Further Related Work

**Fairness.** Several works aim to ensure *fair* predictions of models across subgroups in the data, which are defined based on sensitive demographic information. Commonly used notions of group fairness include demographic parity (Dwork et al., 2012), which aims to ensure that the representation of different demographic groups in the outcomes of a model is similar to their overall representation, and equalized odds (Hardt et al., 2016), which aims for equal predictive performance across groups using metrics such as true positive rate. Fairness interventions used when group information is available include data reweighting to balance groups before training (Kamiran & Calders, 2012), learning separate models for different subgroups (Dwork et al., 2018), and post-processing of trained models, such as adjusting prediction thresholds based on fairness-based metrics (Hardt et al., 2016).

Various approaches have been proposed to achieve fairness when demographic information may not be available. Multicalibration (Hebert-Johnson et al., 2018) aims to learn a model whose predictions are calibrated for all subpopulations that can be identified in a computationally efficient way. Hashimoto et al. (2018) proposes a distributionally robust optimization (DRO)-based approach to minimize the worst-case risk over distributions close to the empirical distribution to ensure fairness. Lahoti et al. (2020) proposes adversarial reweighted learning (ARL), where an auxiliary model identifies subgroups with inferior performance and the model of interest is retrained by reweighting these subgroups to reduce bias.

**Feature Diversification and De-biasing Methods.** Deep neural networks are known to exhibit unwanted biases. For instance, CNNs trained on image data may exhibit texture bias (Geirhos et al., 2022), and language models trained on certain datasets may exhibit annotation bias (Gururangan et al., 2018). Several methods have been proposed to mitigate these biases. Bahng et al. (2020) introduce a framework (ReBias) to

learn de-biased representations by encouraging them to differ from a reference set of biased representations. Li et al. (2022) propose to train two models alternatively, using one to identify biases using an equal opportunity violation criterion and training the other with a reweighted cross-entropy loss to make unbiased predictions. Dagaev et al. (2023) reduce shortcut reliance by upweighting samples based on the misclassification probability of a simple model to train a complex model. Utama et al. (2020a) propose a confidence regularization approach to encourage models to learn from all samples. Recent works also show that deep neural networks tend to amplify the societal biases present in training data (Wang et al., 2019b; Jia et al., 2020) and they propose domain-specific strategies to mitigate such amplification.

Several works aim to improve feature learning for better generalization. Zhang et al. (2022a) aims to learn a shared representation using a succession of training episodes, where they train new classifiers in each episode to do better on misclassified samples from previous episodes. Chen et al. (2023) propose a similar but more efficient approach: at each training step, the data is divided into two sets based on whether the predictions are correct or not, followed by applying DRO on the latter to learn new features with ERM on the former to retain the learned features. Teney et al. (2022) also involves learning a shared representation and training multiple classifiers, where the alignment between their gradients is regularized to encourage feature diversity. Pagliardini et al. (2023); Lee et al. (2022b) do so by encouraging disagreement between two models leveraging unlabeled samples from the target domain.

**Knowledge Distillation.** One can view our approach, in some sense, as the reverse of knowledge distillation (Bucilua et al., 2006; Hinton et al., 2015). These methods aim to transfer knowledge or behaviors from large, complex neural networks into smaller, simpler ones. This can be useful when employing deep learning models in memory-constrained systems such as mobile devices or embedded systems (Gou et al., 2021). To achieve knowledge distillation, Hinton et al. (2015) propose training a small model using a transfer set consisting of samples labeled with the softmax probabilities given by a large pretrained model. Several works have aimed to build upon this approach. Mutual learning (Zhang et al., 2017b) teaches multiple simple models simultaneously, with a KL Divergence-based mimicry loss that ensures the simple models match each others predictions. Teaching assistant (TA) learning (Mirzadeh et al., 2019) first distills knowledge from a large model into an intermediate-sized TA network, then uses the TA network to teach a small model. Comparative knowledge distillation (Wilf et al., 2023), constructs a loss function that teaches a small model to mimic the large model's comparison on two samples from the transfer set, rather than mimic its prediction on individual samples.

While these methods can effectively transfer knowledge without facing significant increases in loss, we do not suspect they would be effective in mitigating simplicity bias. In this paper, we observe that even complex models trained with ERM achieve poor OOD performance due to simplicity bias in deep learning, and using smaller models to mimic their behaviors could further increase the reliance on simple or spurious features due to their limited capacity.

## C  Connections with Prior Work

In general, prior work focuses on one or two of the applications that we consider, while our approach proves effective across several datasets for all cases.

Several methods that aim to improve subgroup robustness or OOD generalization, such as IRM (Arjovsky et al., 2020), GDRO (Sagawa* et al., 2020), PI (Bao et al., 2021) and BLOOD (Bae et al., 2022), require knowledge of group or environment labels. This additional information is not always available and consequently, these methods have limited applicability in such cases. Approaches like JTT (Liu et al., 2021a) and EIIL (Creager et al., 2021) overcome this requirement. However, these methods rely on incorrect predictions (Liu et al., 2021a; Creager et al., 2021) of one model to train another model by upweighting such samples (Liu et al., 2021a) or learning features that generalize across such samples (Creager et al., 2021). As a result, they may not be effective in settings where the spurious feature is highly predictive. Such settings can be relevant when our goal is to mitigate simplicity bias. For example, in the slab dataset (Figure 5), the simple model achieves perfect accuracy but has a much worse margin compared to the global max margin predictor. Our approach is still effective in such cases because it relies on prediction probabilities rather than thresholded predictions,

which enables the differentiation between features that result in smaller margin or low confidence predictions and those that lead to better margin or high confidence predictions.

We note that debiasing/feature diversification methods (Nam et al., 2020; Bahng et al., 2020; Li et al., 2022; Zhang et al., 2022a; Teney et al., 2022) can often be computationally expensive. Some of these involve training two complex models simultaneously (Nam et al., 2020; Bahng et al., 2020) or alternatively (Li et al., 2022). Zhang et al. (2022a); Teney et al. (2022) involves training multiple models and returning the average or selecting the best one among them, respectively. In contrast, our results show that the simple approach to train a single model with the CMI regularization can prove effective across several settings. While methods like Pagliardini et al. (2023); Lee et al. (2022b) are simpler, they require access to additional data from a target domain to encourage disagreement between two models for feature diversification. Our approach does not have such a requirement and allows for diversification directly on the training data.

Several methods use a similar framework of first training a biased model and then training another model debiased or regularized with respect to the first one. We discuss the main differences with these methods here. Many of these methods are designed (or evaluated) specifically for either vision (Jeon et al., 2022; Lee et al., 2022a; Bayasi et al., 2022; Yang et al., 2022; Kim et al., 2022; Taghanaki et al., 2022; Niu et al., 2022; Park et al., 2023; Barbano et al., 2022; Lim et al., 2023) or language (Utama et al., 2020b) datasets, whereas our approach is more broadly applicable to several modalities including vision, language and tabular data. Utama et al. (2020b) is similar to JTT and can suffer from similar drawbacks. Jeon et al. (2022); Bayasi et al. (2022) are designed specifically for CNN-based architectures. Several of these methods are computationally intensive: Lee et al. (2022a); Kim et al. (2022) involve training several biased models before training the final debiased model, Yang et al. (2022) involves training a VAE in the first stage, Taghanaki et al. (2022) involves using an explainability approach to create a masked version of the dataset based on the first stage model to train the final model, Bansal et al. (2022) proposes a model selection strategy that requires training multiple models before selecting the best one, Niu et al. (2022) involves using three models: a pre-trained model, and an autoencoder and the final model that are trained simultaneously, Barbano et al. (2022) involves a three-stage approach, Lim et al. (2023) involves creating adversarial samples for a biased model which are used to train the final model.

In addition, our approach is theoretically grounded and explicit in its assumptions, which could make it easier for a practitioner to evaluate and use. In particular, our definition of spurious features gives an explicit characterization of features/biases that we regard as undesirable and our approach seeks to reduce, which can allow the user to understand the situations when the approach might be effective. We note that most approaches developed to improve subgroup robustness or OOD generalization either lack theoretical guarantees (Bao et al., 2021; Bae et al., 2022; Liu et al., 2021a) or show results tailored to one of these two settings. Sagawa* et al. (2020); Duchi et al. (2019) seek to optimize the worst-case risk over a small set of groups and hence, guarantees on worst-group risk developed in prior work on DRO which considers all subsets of the data as the groups of interest, apply in these cases. Recent work by Deng et al. (2023); Chen et al. (2023) analyzes the learning dynamics of a two-layer CNN in the presence of spurious features. Deng et al. (2023) show that imbalanced data groups and predictive spurious features can cause stronger reliance on these features. Similarly, Chen et al. (2023) show that while the model learns both features, the spurious features are learned faster. However, these methods may not generalize to unseen environments, as is often times the case in the OOD generalization problem. On the other hand, Arjovsky et al. (2020) prove that IRM can recover the optimal invariant predictor that can generalize to any environment, under certain assumptions on the training data. However, follow-up work (Rosenfeld et al., 2021) shows that IRM may fail to recover such a predictor if the training set does not contain samples from several environments. In contrast, we show that our approach reduces reliance on the spurious feature in a Gaussian mixture-based data setting, and also obtain an OOD generalization guarantee in a causal learning framework, without requiring group or environment labels.

## D   Limitations of our Approach

We note that like any other regularization or inductive bias, CMID may not be effective for *every* task. For certain tasks, spurious features as defined by us in Section 2 may not actually be spurious, and we may not always want to reduce reliance on features that are simple and highly predictive of the label. However, it is

worth noting here that it is impossible for a single algorithm to perform well in all cases. This is established by the No Free Lunch theorem: there is no inductive bias that is suitable for all tasks (Shalev-Shwartz & Ben-David, 2014) and relatedly, it is impossible to generalize without certain assumptions on the train and test distributions (Wolpert, 1996).

In most cases, we assume that the test data is i.i.d. and indeed, simplicity bias is useful in explaining in-distribution generalization of NNs in these cases. However, these observations may not necessarily extend to situations where the i.i.d. assumption on the test set is violated. Generalization under distribution shift is significantly different and in such cases, simplicity bias can prevent the model from learning more complex, task-relevant features that may generalize better. Consequently, alleviating simplicity bias can be a useful inductive bias in such cases. This is consistent with our experimental evaluations on several datasets from various modalities and domains.

Even in situations where mitigating simplicity bias is useful, there can be cases where the separation between the feature complexity of spurious and invariant features may not be very large. This can make the selection of an appropriate simple model class for our approach challenging. Indeed, in our experiments, we found that for the CelebA dataset, the spurious and task-relevant features are not significantly different in terms of complexity (Appendix 5.5). Consequently, our approach does not lead to much improvement in the worst-group accuracy on this dataset.

We also note that in general, the use of methods designed to improve OOD generalization and subgroup robustness can lead to a drop in the accuracy on the training set or the i.i.d. test set, compared to ERM. This is seen in some of our experimental results as well (see Fig. 12 and related discussion for more details). This is because a model trained with the ERM objective relies strongly on the spurious feature(s), which is predictive of the train set as well as the i.i.d. test set. In contrast, improvement on OOD test sets is achieved by leveraging other features, which may be less predictive on the train set but generalize better. Some ways to control this trade-off are to change the regularization strength for our approach, or the model selection rule to take both train and validation set accuracies into account. This can allow the user to ensure good training performance, while also attaining improvements in OOD generalization.

# E   Additional Results

## E.1   Effect of CMI Regularization

In this section, we consider the same setup as in Section 4 and present some additional results.

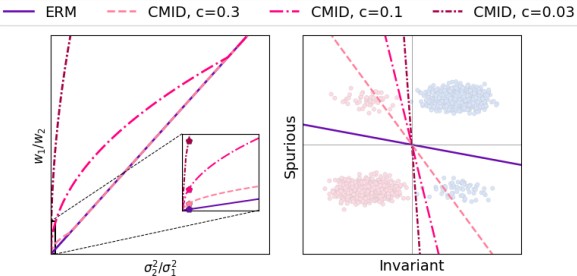

Figure 9: Effect of CMI regularization (wrt $X_2$) for different values of $c$ (corresponding to regularization strength). Left: Lower values of $c$ indicate stronger CMI regularization, resulting in more upweighting of $w_1$ wrt $w_2$. Inset shows a zoomed-in region and markers compare the three solutions when $\sigma_2^2/\sigma_1^2 = 1/6$. Right: Decision boundaries for predictors corresponding to the markers.

In Fig. 9, we visualize the relationship between $\frac{w_1}{w_2}$ and $\frac{\sigma_2^2}{\sigma_1^2}$ predicted in Theorem 1 (assuming $\mu_1 = \mu_2$ and $\eta = 0.95$.) We see that a lower value of $c$ promotes conditional independence with $X_2$ and upweighs $w_1$ more strongly. When $c \to 0$, $w_2 \to 0$.

Next, we consider the case where there are multiple spurious features, by adding another feature

$$X_3 \sim \mathcal{N}(a'\mu_3, \sigma_3^2), \text{ where } a' \sim y\mathcal{R}(\eta') \tag{4}$$

to the setup in Assumption 2. We show that regularizing CMI with respect to the optimal predictor that uses $X_2$ and $X_3$ also results in $w_1$ being upweighted. We consider the constrained problem:

$$\text{ERM}_{\mathcal{C}}(\mathcal{M}) = \underset{\boldsymbol{w} \in \mathcal{M}}{\arg\min} \ \mathbb{E}\left(w_1 X_1 + w_2 X_2 + w_3 X_3 - y\right)^2 \text{ s.t. } I(w_1 X_1 + w_2 X_2 + w_3 X_3; w_2^* X_2 + w_3^* X_3 | y) \leq \nu, \tag{5}$$

where $(w_2^*, w_3^*) \propto \left(\mu_2'/\sigma_2'^2, \mu_3'/\sigma_3'^2\right)$ (the weights for the optimal linear model which uses $X_2$ and $X_3$). We show the following result.

**Theorem 2.** *Let the data be generated as per Assumption 2 and* (4). *Let* $\nu = 0.5\log(1+c^2)$ *for some sufficiently small* $c$, *and* $\frac{\sigma_2^2}{\sigma_2'^2} = \frac{\sigma_3^2}{\sigma_3'^2}$. *Then the solution to* (5) *satisfies:* $\frac{|w_2\mu_2' + w_3\mu_3'|}{|w_1|\sigma_1} = c'$, *where* $c' = 2c\sqrt{\frac{\mu_2'^2}{\sigma_2^2} + \frac{\mu_3'^2}{\sigma_3^2}}$. *Moreover,* $w_2, w_3 \propto c'w_1$.

Similar to before, the result says that for a sufficiently small CMI constraint, we learn a model which mainly uses the invariant feature.

## E.2   Results on the 10-class Colored MNIST Dataset

In this section, we consider the 10-class colored MNIST dataset (following the setup in (Lee et al., 2021)), which is a commonly used dataset for evaluating debiasing methods. In the train set, the samples in every class are injected with a color with 95% correlation, while the colors in the test set are uncorrelated with the label.

In Table 13, we compare the test accuracy of our approach with several debiasing methods, namely LfF (Nam et al., 2020), HEX (Wang et al., 2019a), and recent methods such as EnD (Tartaglione et al., 2021), DFA (Lee et al., 2021) and ReBias (Bahng et al., 2020). We observe that

| Method | HEX | LfF | EnD | DFA | ReBias | CMID |
|---|---|---|---|---|---|---|
| Accuracy | 74.62 | 80.57 | 81.15 | 89.66 | 96.96 | 87.12 |

Table 13: Comparison of OOD test accuracy on 10-class colored MNIST dataset.

CMID significantly improves over HEX, LfF and EnD, and is comparable to DFA, while ReBias significantly outperforms all other methods. We note that ReBias focuses particularly on debiasing with respect to texture or color bias, which is why it is more effective on this dataset.

## E.3   Results using ViT

In this section, we seek to verify that our method remains effective when the complex model is a ViT instead of a CNN. Specifically, we compare the performance of ERM and CMID on the Waterbirds dataset, when training a ViT-B/16 model (Dosovitskiy et al., 2021). As shown in Table 14, our method attains better test performance compared to ERM.

In this experiment, for our approach, we use the same shallow NN as the simple model class, as used in the subgroup robustness experiments. Note that when considering our definition of model complexity (number of trainable parameters, layers, etc.), deep ViTs and CNNs are of the same scale. As a result, the choice of the simple model class can remain the same when considering the two architectures.

| Method | Average Acc | Worst-group Acc |
|---|---|---|
| ERM | 83.8 | 67.3 |
| CMID | 84.8 | 70.1 |

Table 14: Comparison of average and worst-group accuracy on Waterbirds dataset when the complex model is a ViT.

## E.4   Comparison with CLIP

In Table 15, we compare the test performance of CMID with a CLIP ViT-B/16 model (Dosovitskiy et al., 2021), on two datasets. To classify a test image, we first provide CLIP with text descriptions for each class.

For waterbirds, we use ['a landbird', 'a waterbird'], and for CelebA, we use ['a non-blonde', 'a blonde']. We then feed the test image into CLIP and choose the class with the highest cosine similarity to its output as the predicted label.

We note that these datasets are challenging due to imbalances in training data. For example, predicting "blonde" vs. "not-blonde" on the CelebA dataset is difficult because there are very few blonde males in the training data. The imbalances in CLIP training data, on the other hand, are unclear. While CLIP models may perform well on some benchmarks, it remains uncertain if they are effective in all cases.

| Dataset | Method | Average Acc | Worst-group Acc |
|---|---|---|---|
| Waterbirds | CLIP | 78.3 | 29.3 |
| | CMID | 88.6 | 84.3 |
| CelebA | CLIP | 88.7 | 40.4 |
| | CMID | 84.5 | 75.3 |

Table 15: Comparison of average and worst-group accuracy on Waterbirds and CelebA datasets with CLIP.

The results above demonstrate that CLIP models can be susceptible to natural spurious correlations, highlighting the need for grounded and consistent robustness techniques, such as the proposed method CMID.

## F   Proofs for Theoretical Results

In this section, we present the proofs for the theoretical results in Section 4 and Appendix E.

### F.1   Proof of Proposition 1

We first restate Proposition 1:

**Proposition 1.** *For all $i \in [d]$, ERM($\mathcal{M}$) satisfies $\frac{w_i}{w_{d+1}} = \frac{\mu_1}{\mu_2'} \frac{\sigma_2'^2}{\sigma_1^2}$. When $\frac{\mu_1}{\mu_2'} \frac{\sigma_2'^2}{\sigma_1^2} < 1$, ERM($\mathcal{M}_s$)$= \left[ 0, \ldots, 0, \frac{\mu_2'}{\sigma_2^2} \right]$ (upto scaling).*

*Proof.* We have:

$$\mathbb{E} \left( \sum_{i=1}^{d+1} w_i X_i - y \right)^2 = (\sigma_1^2 + \mu_1^2) \sum_{i=1}^{d} w_i^2 + w_{d+1}^2 (\sigma_2^2 + \mu_2^2) + \sum_{i=1}^{d} \sum_{j=i+1}^{d} 2 w_i w_j \mu_1^2 + \sum_{i=1}^{d} 2 w_i w_{d+1} \mu_1 \mu_2'$$

$$- 2\mu_1 \sum_{i=1}^{d} w_i - 2 w_{d+1} \mu_2' + 1.$$

Since $\underset{\boldsymbol{w} \in \mathcal{M}}{\arg \min} \mathbb{E} \left( \sum_{i=1}^{d+1} w_i X_i - y \right)^2$ is a convex problem, to find the minimizer we set gradients with respect to $\boldsymbol{w}$ to be 0. Subsequently, we solve the resulting set of equations. By taking the gradient and setting it to 0, we obtain the following set of equations:

$$2 w_i (\sigma_1^2 + \mu_1^2) - 2\mu_1 + 2\mu_1^2 \sum_{j=1, j \neq i}^{d} w_j + 2 w_{d+1} \mu_1 \mu_2' = 0 \text{ for } i \in [d], \tag{6}$$

$$2 w_{d+1} (\sigma_2^2 + \mu_2^2) - 2\mu_2' + 2\mu_1 \mu_2' \sum_{j=1}^{d} w_j = 0. \tag{7}$$

From (6), we have that $w_i = w_j$ for all pairs $i, j \in [d]$ such that $i \neq j$. Specifically $w_i = \dfrac{\mu_1 - w_{d+1} \mu_1 \mu_2'}{d\mu_1^2 + \sigma_1^2}$ for all $i \in [d]$. Substituting this into (7) and solving for $w_{d+1}$, we get:

$$w_{d+1} = \frac{\mu_2'}{\sigma_2'^2} \left( \frac{1}{\frac{d\mu_1^2}{\sigma_1^2} + \frac{\mu_2'^2}{\sigma_2^2} + 1} \right),$$

where $\sigma_2'^2 = \sigma_2^2 + \mu_2^2 - \mu_2'^2$. Using this, we get the expression for $w_i$, when $i \in [d]$:

$$w_i = \frac{\mu_1}{\sigma_1^2} \left( \frac{1}{\frac{d\mu_1^2}{\sigma_1^2} + \frac{\mu_2'^2}{\sigma_2'^2} + 1} \right).$$

Thus, for ERM($\mathcal{M}$) and any $i \in [d]$, $\frac{w_i}{w_{d+1}} = \frac{\mu_1}{\mu_2'} \frac{\sigma_2'^2}{\sigma_1^2}$.

Next, consider ERM over $\mathcal{M}_s$. If for some $i \in [d]$, $w_i$ is non-zero, we use (6) and get $\left[ 0, \ldots, \frac{\mu_1}{\sigma_1^2 + \mu_1^2}, \ldots, 0 \right]$ as the solution, for which the loss value is $\frac{\sigma_1^2}{\sigma_1^2 + \mu_1^2}$. If $w_i = 0$ for all $i \in [d]$, we use (7) and get $\left[ 0, \ldots, 0, \frac{\mu_2'}{\sigma_2'^2 + \mu_2'^2} \right]$ as the solution, for which the loss is $\frac{\sigma_2'^2}{\sigma_2'^2 + \mu_2'^2}$. When $\frac{\mu_1^2}{\mu_2'^2} \frac{\sigma_2'^2}{\sigma_1^2} < 1$, the latter has a smaller loss and thus, ERM($\mathcal{M}_s$) is $\left[ 0, \ldots, 0, \frac{\mu_2'}{\sigma_2'^2} \frac{1}{1 + \frac{\mu_2^2}{\sigma_2^2}} \right]$. $\qquad\square$

### F.2   Proof of Theorem 1

**Theorem 1.** *Let data be generated as per Assumption 2. For $\nu = 0.5 \log(1 + c^2)$ for some c:*

*1. When $\frac{\mu_1}{\mu_2'} \frac{\sigma_2'^2}{\sigma_1 \sigma_2} > \frac{1}{c\sqrt{d}}$, the solution to (1) is the same as ERM($\mathcal{M}$), so for $i \in [d]$, $\frac{w_i}{w_{d+1}} = \frac{\mu_1}{\mu_2'} \frac{\sigma_2'^2}{\sigma_1^2}$.*

*2. Otherwise, for $i \in [d]$, $w_i$ is upweighted and the solution to (1) satisfies $\frac{|w_i|}{|w_{d+1}|} = \frac{1}{c\sqrt{d}} \frac{\sigma_2}{\sigma_1}$.*

*Proof.* Consider the constraint $I\left( \sum_{i=1}^{d+1} w_i X_i; w_{d+1}^* X_{d+1} | y \right) \leq \nu$. As we are working with continuous random variables in this setup, we employ differential entropy for our entropy computations. The entropy of a Gaussian random variable $X$ with variance $\sigma^2$ is given by $H(X) = 0.5(\log(2\pi\sigma^2) + 1)$. Using this and the definitions of CMI and conditional entropy, we have:

$$I\left( \sum_{i=1}^{d+1} w_i X_i; w_{d+1}^* X_{d+1} | y \right) = H\left( \sum_{i=1}^{d+1} w_i X_i | y \right) - H\left( \sum_{i=1}^{d+1} w_i X_i | y, w_{d+1}^* X_{d+1} \right)$$

$$= H\left( \sum_{i=1}^{d+1} w_i X_i | y \right) - H\left( \sum_{i=1}^{d} w_i X_i | y \right) = \frac{1}{2} \log\left( \frac{\sigma_1^2 \|\boldsymbol{w}_I\|_2^2 + \sigma_2^2 w_{d+1}^2}{\sigma_1^2 \|\boldsymbol{w}_I\|_2^2} \right).$$

where $\boldsymbol{w_I} = (w_1, \ldots, w_d)$. Using $\nu = 0.5 \log(1 + c^2)$, the constraint becomes: $\frac{w_{d+1}^2 \sigma_2^2}{\|\boldsymbol{w}_I\|_2^2 \sigma_1^2} \leq c^2$. Thus, (1) reduces to solving:

$$\min_{\boldsymbol{w}} \mathbb{E} \left( \sum_{i=1}^{d+1} w_i X_i - y \right)^2 \quad \text{s.t.} \quad \frac{|w_{d+1}| \sigma_2}{\|\boldsymbol{w}_I\|_2 \sigma_1} \leq c.$$

If $\frac{\mu_2'}{\mu_1} \frac{\sigma_1 \sigma_2}{\sigma_2'^2} \leq c\sqrt{d}$, ERM($\mathcal{M}$) satisfies the constraint and serves as the solution to (1). Otherwise, since this is a convex optimization problem with an affine constraint, the constraint must be tight. Therefore, we determine the solution by finding ERM($\mathcal{M}$) subject to $\frac{|w_{d+1}| \sigma_2}{\|\boldsymbol{w}_I\|_2 \sigma_1} = c$. The solution, for $i \in [d]$ is given by:

$$w_i = \frac{\mu_1}{\sigma_1^2} \left( \frac{1}{1 + \frac{d\mu_1^2}{\sigma_1^2} \pm \sqrt{d}\mu_1 \mu_2' \frac{c}{\sigma_1 \sigma_2}} \right), \quad w_{d+1} = \pm \frac{c\sqrt{d}\sigma_1 \mu_1}{\sigma_2 \sigma_1^2} \left( \frac{1}{1 + \frac{d\mu_1^2}{\sigma_1^2} \pm \sqrt{d}\mu_1 \mu_2' \frac{c}{\sigma_1 \sigma_2}} \right).$$

$\qquad\square$

### F.3 Proof of Theorem 2

**Theorem 2.** *Let the data be generated as per Assumption 2 and (4), with $d = 1$. Let $\nu = 0.5 \log(1 + c^2)$ for some sufficiently small $c$, and $\frac{\sigma_2^2}{\sigma_2'^2} = \frac{\sigma_3^2}{\sigma_3'^2}$. Then the solution to (5) satisfies: $\frac{|w_2\mu_2' + w_3\mu_3'|}{|w_1|\sigma_1} = c'$, where $c' = 2c\sqrt{\frac{\mu_2'^2}{\sigma_2^2} + \frac{\mu_3'^2}{\sigma_3^2}}$. Moreover, $w_2, w_3 \propto c'w_1$.*

*Proof.* Let $Z_1 = w_1 X_1 + w_2 X_2 + w_3 X_3$ and $Z_2 = w_2^* X_2 + w_3^* X_3$. Then,

$$I(Z_1, Z_2 | y) = H(Z_1 | y) + H(Z_2 | y) - H(Z_1, Z_2 | y).$$

Since all features are Gaussian, we have:

$$H(Z_1 | y) = 0.5(\log(w_1^2 \sigma_1^2 + w_2^2 \sigma_2^2 + w_3^2 \sigma_3^2) + \log(2\pi) + 1),$$
$$H(Z_2 | y) = 0.5(\log((w_2^*)^2 \sigma_2^2 + (w_3^*)^2 \sigma_3^2) + \log(2\pi) + 1),$$
$$H(Z_1, Z_2 | y) = 0.5 \log |K| + \log(2\pi) + 1,$$

where $K$ is the covariance matrix of $Z_1$ and $Z_2$ conditioned on $y$. We can calculate $K$ as follows. Since all features are conditionally independent, we have:

- $\mathbb{E}(Z_1 - \mathbb{E}(Z_1))^2 = w_1^2 \sigma_1^2 + w_2^2 \sigma_2^2 + w_3^2 \sigma_3^2$.

- $\mathbb{E}(Z_1 - \mathbb{E}(Z_1))(Z_2 - \mathbb{E}(Z_2)) = w_2 w_2^* \sigma_2^2 + w_3 w_3^* \sigma_3^2$.

- $\mathbb{E}(Z_2 - \mathbb{E}(Z_2))^2 = (w_2^*)^2 \sigma_2^2 + (w_3^*)^2 \sigma_3^2$.

Thus, $|K| = (w_1^2 \sigma_1^2 + w_2^2 \sigma_2^2 + w_3^2 \sigma_3^2)((w_2^*)^2 \sigma_2^2 + (w_3^*)^2 \sigma_3^2) - (w_2 w_2^* \sigma_2^2 + w_3 w_3^* \sigma_3^2)^2$. Using these, the constraint becomes:

$$\log\left(\frac{(w_1^2 \sigma_1^2 + w_2^2 \sigma_2^2 + w_3^2 \sigma_3^2)((w_2^*)^2 \sigma_2^2 + (w_3^*)^2 \sigma_3^2)}{(w_1^2 \sigma_1^2 + w_2^2 \sigma_2^2 + w_3^2 \sigma_3^2)((w_2^*)^2 \sigma_2^2 + (w_3^*)^2 \sigma_3^2) - (w_2 w_2^* \sigma_2^2 + w_3 w_3^* \sigma_3^2)^2}\right) \le \log(1 + c^2)$$

$$\implies \frac{(w_2 w_2^* \sigma_2^2 + w_3 w_3^* \sigma_3^2)^2}{w_1^2 \sigma_1^2((w_2^*)^2 \sigma_2^2 + (w_3^*)^2 \sigma_3^2) + \sigma_2^2 \sigma_3^2 (w_2^* w_3 - w_3^* w_2)^2} \le c^2$$

$$\implies \frac{(w_2 w_2^* \sigma_2^2 + w_3 w_3^* \sigma_3^2)^2 - c^2 \sigma_2^2 \sigma_3^2 (w_2^* w_3 - w_3^* w_2)^2}{w_1^2 \sigma_1^2((w_2^*)^2 \sigma_2^2 + (w_3^*)^2 \sigma_3^2)} \le c^2.$$

Assume that $c$ is sufficiently small, *i.e.*, $c \le \frac{\sqrt{3}|w_2 w_2^* \sigma_2^2 + w_3 w_3^* \sigma_3^2|}{2\sigma_2 \sigma_3 |w_2 w_3^* - w_3 w_2^*|}$. Then, we get the condition:

$$\frac{(w_2 w_2^* \sigma_2^2 + w_3 w_3^* \sigma_3^2)^2}{w_1^2 \sigma_1^2((w_2^*)^2 \sigma_2^2 + (w_3^*)^2 \sigma_3^2)} \le 4c^2$$

$$\implies \frac{\left| w_2 \mu_2' \frac{w_2^* \sigma_2^2}{\mu_2'} + w_3 \mu_3' \frac{w_3^* \sigma_3^2}{\mu_3'} \right|}{|w_1|\sigma_1} \le 2c\sqrt{(w_2^*)^2 \sigma_2^2 + (w_3^*)^2 \sigma_3^2}.$$

When $[w_2^*, w_3^*] = v\left[\mu_2'/\sigma_2'^2, \mu_3'/\sigma_3'^2\right]$, where $v$ is some constant, we get:

$$\frac{\left| w_2 \mu_2' \frac{\sigma_2^2}{\sigma_2'^2} + w_3 \mu_3' \frac{\sigma_3^2}{\sigma_3'^2} \right|}{|w_1|\sigma_1} \le 2c\sqrt{\frac{\mu_2'^2 \sigma_2^2}{\sigma_2'^4} + \frac{\mu_3'^2 \sigma_3^2}{\sigma_3'^4}}.$$

Since $\frac{\sigma_2^2}{\sigma_2'^2} = \frac{\sigma_3^2}{\sigma_3'^2}$, we get:

$$\frac{|w_2 \mu_2' + w_3 \mu_3'|}{|w_1|\sigma_1} \le 2c\sqrt{\frac{\mu_2'^2}{\sigma_2^2} + \frac{\mu_3'^2}{\sigma_3^2}} = c'.$$

When $\frac{\frac{\mu_2'^2}{\sigma_2'^2} + \frac{\mu_3'^2}{\sigma_3'^2}}{\frac{\mu_1}{\sigma_1}} \le c'$, ERM($\mathcal{M}$), *i.e.*, $[w_1, w_2, w_3] \propto \left[\frac{\mu_1}{\sigma_1^2}, \frac{\mu_2'}{\sigma_2'^2}, \frac{\mu_3'}{\sigma_3'^2}\right]$ satisfies the constraint, and thus is the solution to (5). Otherwise, since (5) is a convex problem with an affine constraint, the constraint must be

tight. Therefore, we find $\text{ERM}(\mathcal{M})$ subject to the equality constraint $|w_2\mu_2' + w_3\mu_3'| = c'\sigma_1|w_1|$. The solution is given by:

$$w_1 = \frac{\mu_1}{\sigma_1^2} \frac{1+c'\frac{\sigma_1}{\mu_1}}{1+\frac{\mu_1^2}{\sigma_1^2}+2c'\frac{\mu_1}{\sigma_1}+(c')^2\frac{1-\frac{\mu_2'^2}{\sigma_2^2+\mu_2^2}\frac{\mu_3'^2}{\sigma_3^2+\mu_3^2}}{\frac{\mu_2'^2}{\sigma_2^2+\mu_2^2}+\frac{\mu_3'^2}{\sigma_3^2+\mu_3^2}-2\frac{\mu_2'^2}{\sigma_2^2+\mu_2^2}\frac{\mu_3'^2}{\sigma_3^2+\mu_3^2}}},$$

$$w_2 = c'\sigma_1 w_1 \frac{\frac{\mu_2'}{\sigma_2^2+\mu_2^2}\left(1-\frac{\mu_3'^2}{\sigma_3^2+\mu_3^2}\right)}{\frac{\mu_2'^2}{\sigma_2^2+\mu_2^2}+\frac{\mu_3'^2}{\sigma_3^2+\mu_3^2}-2\frac{\mu_2'^2}{\sigma_2^2+\mu_2^2}\frac{\mu_3'^2}{\sigma_3^2+\mu_3^2}}, \quad w_3 = c'\sigma_1 w_1 \frac{\frac{\mu_3'}{\sigma_3^2+\mu_3^2}\left(1-\frac{\mu_2'^2}{\sigma_2^2+\mu_2^2}\right)}{\frac{\mu_2'^2}{\sigma_2^2+\mu_2^2}+\frac{\mu_3'^2}{\sigma_3^2+\mu_3^2}-2\frac{\mu_2'^2}{\sigma_2^2+\mu_2^2}\frac{\mu_3'^2}{\sigma_3^2+\mu_3^2}}.$$

$\square$

### F.4 Proof of Proposition 2

**Proposition 2.** *Let $\text{ERM}(\mathcal{M}_s) = M_s^*$. Under Assumptions 3 and 4, the solution to the problem:*

$$\underset{M\in\mathcal{M}}{\arg\min}\, \mathbb{E}\,\ell_M(Z) \ \text{s.t.}\ I(M; M_s^*|Y) = 0 \tag{2}$$

*is $M = \Phi^*$, the maximal invariant predictor.*

*Proof.* Using Assumption 3, the class of simple models only contains variant predictors, so $\text{ERM}(\mathcal{M}_s) = \Psi_s$. Consequently, the constraint in (2) can be written as $I(M; \Psi_s|Y) = 0$.

Considering the set of candidate predictors for $M$, namely $\{0, \Phi_c, \Psi_s, \Psi_c\}$, we examine the CMI constraint for each. Using the definition of mutual information, we have $I(0, \Psi_s|Y) = 0$ and $I(\Psi_s, \Psi_s|Y) = H(\Psi_s|Y)$. According to the definition of *variant* predictor, $H(\Psi_s|Y) > H(\Psi_s|Y, E) \geq 0$.

From Assumption 4, which states that the invariant and variant predictors are conditionally independent, we can deduce that $I(\Phi_c, \Psi_s|Y) = 0$. From Assumption 4, we also have $I(\Psi_c, \Psi_s|Y) > I(\Psi_c, \Psi_s|Y, E) \geq 0$.

Using these results, the feasible set is $[0, \Phi_c]$, which corresponds to the invariance set $\mathcal{I}_E(\mathcal{M})$. Consequently, problem (2) is equivalent to finding $\underset{\Phi\in\mathcal{I}_E(\mathcal{M})}{\arg\max} I(Y; \Phi)$. The solution to this problem is $\Phi_c$, which represents the MIP $\Phi^*$. $\square$

## G   Experimental Settings

We begin by describing some common details and notation that we use throughout this section. As in the main text, we use $\lambda$ to represent the regularization strength for CMI. To ensure effective regularization, we adopt an epoch-dependent approach by scaling the regularization strength using the parameter $S$. Specifically, we set $\lambda = \lambda_c (1 + t/S)$ at epoch $t$. The temperature parameter $T$ is set as 12.5 throughout the experiments. Additionally, we use LR to denote the learning rate, BS to denote the batch size, and $\lambda_2$ to denote the weight decay parameter, which represents the strength of $\ell_2$-regularization. When using the Adam optimizer, we employ the default values for momentum.

The experiments on Slab data, CMNIST and CPMNIST data and Adult-Confounded data were implemented on Google Colab. The ImageNet-9 experiments were run on an AWS G4dn instance with one NVIDIA T4 GPU. For experiments on the subgroup robustness datasets and the Camelyon17-WILDS data, we used two NVIDIA V100 GPUs with 32 GB memory each. We only used CPU cores for the Bios data experiments.

### G.1   Mitigating Simplicity Bias Experiments

This section includes the details for the experiments showing that CMID mitigates simplicity bias where we use the Slab data and the ImageNet-9 data.

### G.1.1 Slab Data

**Dataset.** All the features in the 3-Slab and 5-Slab data are in the range $[-1, 1]$. The features are generated by defining the range of the slabs along each direction and then sampling points in that range uniformly at random. The base code for data generation came from the official implementation of Shah et al. (2020) available at `https://github.com/harshays/simplicitybiaspitfalls`. We consider $10^5$ training samples and $5 \times 10^4$ test samples. In both cases, the linear margin is set as 0.05. The 3-Slab data is 10-dimensional, where the remaining 8 coordinates are standard Gaussians and are not predictive of the label. The slab margin is set as 0.075. The 5-slab data is only 2-dimensional, and the slab margin is set as 0.14.

**Training.** We consider a linear model for the simple model and following Shah et al. (2020), a 1-hidden layer NN with 100 hidden units as the final model (for both ERM and CMID). Throughout, we use SGD with BS $= 500, \lambda_2 = 5 \times 10^{-4}$ for training. The linear model is trained with LR $= 0.05$, while the NN is trained with LR $= 0.005$.

For the 3-Slab data, the models are trained for 300 epochs. We consider $\lambda_c \in \{100, 150, 200\}$ and choose $\lambda_c = 150$ for the final result. For the 5-Slab data, the models are trained for 200 epochs and we use a 0.99 momentum in this case. We consider $\lambda_c \in \{1000, 2000, 2500, 3000\}$ and choose $\lambda_c = 3000$ for the final result. Note that we consider significantly high values of $\lambda_c$ for this dataset compared to the rest because the simple model is perfectly predictive of the label in this case. This implies that its CMI with the final model is very small, and the regularization strength needs to be large in order for this term to contribute to the loss.

### G.1.2 Texture vs Shape Bias on ImageNet-9

**Dataset** ImageNet-9 (Xiao et al., 2020) is a subset of ImageNet with nine condensed classes that each consist of images from multiple ImageNet classes. These include dog, bird, wheeled vehicle, reptile, carnivore, insect, musical instrument, primate, and fish. ImageNet-A (Hendrycks et al., 2021) is a set of handpicked images that ResNet50 models trained on ImageNet classify incorrectly. We organize these images into the same nine classes as ImageNet-9.

**Calculating shape bias** The GST dataset (Geirhos et al., 2022) consists of 16 shape classes. To interpret a prediction from a model trained on ImageNet-9 as a GST dataset prediction, we consider a subset of classes from both and use the mapping listed in Table 16. Specifically, to determine whether a model trained on ImageNet-9 predicts correctly on a GST image, we first determine which of the 5 ImageNet-9 classes from the Table has the highest probability based on the model's output, and then use the mapping to obtain the predicted GST class label.

| ImageNet-9 Class | GST Dataset Classes |
|---|---|
| dog | dog |
| bird | bird |
| wheeled vehicle | bicycle, car, truck |
| carnivore | bear, cat |
| musical instrument | keyboard |

Table 16: ImageNet-9 classes mapped to corresponding GST dataset classes.

Thus, following the procedure detailed in `https://github.com/rgeirhos/texture-vs-shape`, the shape bias is calculated as:

$$\text{shape bias} = \frac{\text{number of correct shape predictions}}{\text{number of correct shape predictions} + \text{number of correct texture predictions}}.$$

**Training** We use ResNet18 pre-trained on ImageNet data as the simple model and train it on ImageNet-9 using Adam with LR $= 0.001, \text{BS} = 32, \lambda_2 = 10^{-4}$ for 2 epochs. We do not consider a simpler architecture and training from scratch since this pre-trained model already exhibits texture bias. For the final model, we consider the same model and batch size, but we use SGD with 0.9 momentum as the optimizer, LR $= 10^{-4}, \lambda_2 = 0.001$, and train for 10 epochs. We report the mean and standard deviation over 3 runs, where for each run we select the model with the highest shape bias. Values of CMID-specific parameters were $\lambda_c = 30, S = 10$. For tuning, we consider LR $\in \{10^{-5}, 10^{-4}, 10^{-3}\}$ for both the models and BS $\in \{16, 32\}, \lambda_2 \in \{0.0001, 0.001, 0.01\}$ and $\lambda_c \in \{0.5, 15, 30, 50\}$. We implement JTT to obtain the results. We consider the upsampling parameter (Liu et al., 2021a) $\lambda_{up} \in \{5, 20, 50, 100\}$ for tuning. The other hyperparameters are the same as our approach.

## G.2 Better OOD Generalization Experiments

This section includes the details of the experiments showing that CMID leads to better OOD generalization. For this, we used CMNIST and CPMNIST, Camelyon17-WILDS and Adult-Confounded datasets.

### G.2.1 CMNIST and CPMNIST

**Dataset.** Following Bae et al. (2022), we use $25,000$ MNIST images (from the official train split) for each of the training environments, and the remaining $10,000$ images to construct a validation set. For both test sets, we use the $10,000$ images from the official test split.

**Training.** The details about the model architecture and parameters for training the simple model with ERM and the final model with CMID, for both datasets, are listed in Table 17. Following Bae et al. (2022), the MLP has one hidden layer with 390 units and ReLU activation function. In both cases, the simple model is trained for 4 epochs, while the final model is trained for 20 epochs. We choose the model with the smallest accuracy gap between the training and validation sets. For tuning, we consider the following values for each parameter: for the simple model, LR $\in \{0.005, 0.01, 0.05\}$, $\lambda_2 \in \{0.001, 0.005, 0.01\}$, and for the final model, LR $\in \{0.001, 0.005\}$, $\lambda_c \in [3, 8]$ and $S \in [3, 6]$, where lower values of $S$ were tried for higher values of $\lambda_c$ and vice-versa.

For the final results, we report the mean and standard deviation by averaging over 4 runs. For comparison, we consider the results reported by Bae et al. (2022) for all methods, except EIIL (Creager et al., 2021) and JTT (Liu et al., 2021a). Results for EIIL are obtained by using their publicly available implementation for CMNIST data (available at `https://github.com/ecreager/eiil`) and incorporating the CPMNIST data into their implementation. The hyperparameter values in their implementation are kept the same. We implement JTT to obtain the results. We consider LR $\in \{0.001, 0.005, 0.01\}$ and the parameter for upweighting minority groups (Liu et al., 2021a) $\lambda_{up} \in \{5, 10, 15, 20, 25\}$ for tuning.

| Dataset | Simple Model | Optimizer | LR | BS | $\lambda_2$ | Final Model | Optimizer | LR | BS | $\lambda_c$ | $S$ |
|---------|--------------|-----------|------|----|-------------|-------------|-----------|-------|----|-------------|-----|
| CMNIST | Linear | SGD | 0.01 | 64 | 0.005 | MLP | SGD | 0.001 | 64 | 4 | 4 |
| CPMNIST | Linear | SGD | 0.01 | 64 | 0.001 | MLP | SGD | 0.005 | 64 | 5 | 3 |

Table 17: Training details for CMNIST and CPMNIST.

### G.2.2 Camelyon17-WILDS

**Dataset.** Camelyon17-WILDS (Koh et al., 2021) contains $96 \times 96$ image patches which may or may not display tumor tissue in the central region. We use the same dataset as Bae et al. (2022), which includes $302,436$ training patches, $34,904$ OOD validation patches, and $85,054$ OOD test patches, where no two data splits contain images from overlapping hospitals. We use the WILDS package, available at `https://github.com/p-lambda/wilds` for data-loading.

**Training.** For the simple model, we train a `2DConvNet1` model (see Section G.3.2 for details) for 10 epochs. We use the Adam optimizer with LR $= 10^{-4}$, BS $= 32$, $\lambda_2 = 10^{-4}$. For the final model, we train a DenseNet121 (randomly initialized, no pretraining) for 5 epochs using SGD with 0.9 momentum with LR $= 10^{-4}$, BS $= 32$, $\lambda_2 = 0.01$ and $\lambda_c = 0.5$, $S = 10$. We use the same BS and $\lambda_2$ values as Bae et al. (2022) for consistency. For tuning, we consider LR $\in \{10^{-5}, 10^{-4}, 10^{-3}\}$ for both the models and $\lambda_c \in \{0.5, 2, 5, 15\}$ for CMID. While Bae et al. (2022) and Koh et al. (2021) use learning rates $10^{-5}$ and $0.001$, respectively, we found a learning rate of $10^{-4}$ was most suited for our approach. We select the model with the highest average accuracy on the validation set to report the final results and report the mean and standard deviation by averaging over 3 runs. For comparison, we use the results reported by Bae et al. (2022) for all the methods, except JTT. For the results for JTT, we implement the method and tune $\lambda_{up} \in \{20, 50, 100\}$ and the number of epochs to train the first-stage model as $\{1, 2\}$. The rest of the hyperparameters are kept the same as our approach. We noticed that JTT (Liu et al., 2021a) had a high variance in test accuracy across multiple runs for different random weight initialization. To account for this, we ran JTT and CMID over the same 8 seeds and randomly chose 3 of them to report the average accuracies over.

### G.2.3 Adult-Confounded

**Dataset.** The UCI Adult dataset (Newman et al., 1998; Leisch & Dimitriadou, 2021), comprises $48,842$ census records collected from the USA in 1994. It contains attributes based on demographics and employment information and the target label is a binarized income indicator (thresholded at \$50,000). The task is commonly used as an algorithmic fairness benchmark. Lahoti et al. (2020); Creager et al. (2021) define four sensitive subgroups based on binarized sex (Male/Female) and race (Black/non-Black) labels: Non-Black Males (G1), Non-Black Females (G2), Black Males (G3), and Black Females (G4). They observe that each subgroup has a different correlation strength with the target label ($p(y = 1|G)$), and thus, in some cases, subgroup membership alone can be used to achieve a low error rate in prediction.

Based on this observation, Creager et al. (2021) create a semi-synthetic variant of this data, known as Adult-Confounded, where they exaggerate the spurious correlations in the original data. As G1 and G3 have higher values of $p(y = 1|G)$ across both the splits, compared to the other subgroups (see Creager et al. (2021) for exact values), these values are increased to 0.94, while they are set to 0.06 for the remaining two subgroups, to generate the Adult-Confounded dataset. In the test set, these are reversed, so that it serves as a worst-case audit to ensure that the model is not relying on subgroup membership alone in its predictions. Following Creager et al. (2021), we generate the samples for the Adult-Confounded dataset by using importance sampling. We use the original train/test splits from UCI Adult as well as the same subgroup sizes, but individual examples are under/over-sampled using importance weights based on the correlation between the original data and the desired correlation.

**Training.** We use a linear model (with a bias term) as the simple model, and following Creager et al. (2021) use an Adagrad optimizer throughout. We use BS $= 50$. The simple model is trained for 50 epochs, with LR $= 0.05, \lambda_2 = 0.001$. Following Creager et al. (2021); Lahoti et al. (2020), we use a two-hidden-layer MLP architecture for the final model, with 64 and 32 hidden units, respectively. It is trained with LR $= 0.04, \lambda_c = 4, S = 4$ for 10 epochs. We also construct a small validation set from the train split by randomly selecting a small fraction of samples ($5 - 50$) from each subgroup (depending on its size) and then upsampling these samples to get balanced subgroups of size 50. We choose the model with the lowest accuracy gap between the train and validation sets. For tuning, we consider LR $\in \{0.01, 0.02, 0.03, 0.04, 0.05\}$ and $\lambda_c, S \in \{3, 4, 5\}$. For the final results, we report the mean and accuracy by averaging over 4 runs. For comparison, we reproduced the results for ERM from Creager et al. (2021), and thus, consider the values reported in Creager et al. (2021) for ERM, ARL and EIIL. We implement JTT to obtain the results, using $LR = 0.05$ and $\lambda_2 = 0.001$ to train the final model. We tune the parameter for upweighting minority groups (Liu et al., 2021a) $\lambda_{up}$ over $\{10, 20, 25, 30\}$. The remaining parameters are kept the same as for our approach.

### G.3 Subgroup Robustness Experiments

This section includes the details of the experiments showing that CMID enhances subgroup robustness, where we use four benchmark datasets: Waterbirds, CelebA, MultiNLI and CivilComments.

### G.3.1 Datasets

We consider the following datasets for this task. We follow the setup in Sagawa* et al. (2020) for the first three and Koh et al. (2021) for CivilComments-WILDS.

**Waterbirds** Waterbirds is a synthetic dataset created by Sagawa* et al. (2020) consisting of bird images over backgrounds. The task is to classify whether a bird is a *landbird* or a *waterbird*. The background of the image *land background* or *water background*, acts a spurious correlation.

**CelebA** CelebA is a synthetic dataset created by Liu et al. (2015) containing images of celebrity faces. We classify the hair color as *blonde* or *not blonde*, which is spuriously correlated with the gender of the celebrity *male* or *female*, as done in Sagawa* et al. (2020); Liu et al. (2021a).

**MultiNLI** MultiNLI (Williams et al., 2018) is a dataset of sentence pairs consisting of three classes: entailment, neutral, and contradiction. Pairs are labeled based on whether the second sentence entails, is

neutral with, or contradicts the first sentence, which is correlated with the presence of negation words in the second sentence (Sagawa* et al., 2020; Liu et al., 2021a).

**CivilComments-WILDS** CivilComments-WILDS is a dataset of online comments proposed by Borkan et al. (2019). The goal is to classify whether a comment is *toxic* or *non-toxic*, which is spuriously correlated with the mention of one or more of the following demographic attributes: male, female, White, Black, LGBTQ, Muslim, Christian, and other religion (Park et al., 2018; Dixon et al., 2018). Similar to previous work (Koh et al., 2021; Liu et al., 2021a), we evaluate over 16 overlapping groups, one for each potential label-demographic pair.

### G.3.2 Model Architectures

In this section, we discuss the architectures we consider for the simple models for this task. For the two image datasets, a shallow 2D CNN is a natural choice for the simple model as 2D CNNs can capture local patterns and spatial dependencies in grid-like data. On the other hand, for the two text datasets with tokenized representations, we consider a shallow MLP or 1D CNN for the simple model. MLPs can capture high-level relationships between tokens by treating each token as a separate feature, while 1D CNNs can capture local patterns and dependencies in sequential data.

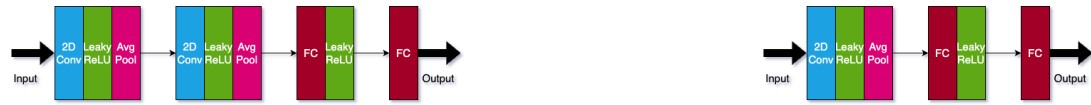

Figure 10: Left: `2DConvNet1` and Right: `2DConvNet2` architectures.

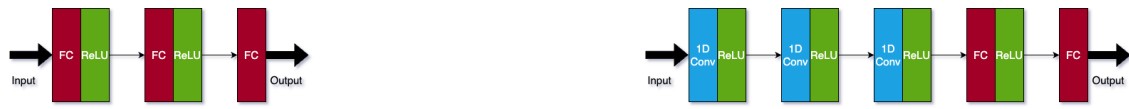

Figure 11: Left: `2MLP` and Right: `1DConvNet` architectures.

Next, we describe the details for the model architectures. Let $F$ denote the filter size and $C$ denote the number of output channels (for convolutional layers) or the output dimension (for linear/fully connected (FC) layers). Throughout, we use $F=2$ for the average pooling layers. Fig. 10 shows the `2DConvNet1` and the `2DConvNet2` architecture, which were used as simple models for Waterbirds and CelebA, respectively. These were the only two architectures we considered for the 2DCNN on these datasets. In `2DConvNet1`, the 2D convolutional layers use $F=7, C=10$ and $F=4, C=20$, respectively, while $C=2000$ for the FC layer. In the `2DConvNet2` architecture, $F=5, C=10$ for the 2D convolutional layer and $C=500$ for the FC layer. Fig. 11 shows the `2MLP` and the `1DConvNet` architecture, which were used as simple models for MultiNLI and CivilComments-WILDS, respectively. For tuning, we considered these models as well as a 1DCNN with one less 1D convolutional layer than `1DConvNet` for both datasets. In `2MLP`, the FC layers use $C=100$ and $C=25$, respectively. In the `1DConvNet` architecture, the 1D convolutional layers use $F=7, C=10$, $F=5, C=32$ and $F=5, C=64$, respectively, while $C=500$ for the FC layer.

### G.3.3 Training Details

We utilize the official implementation of Sagawa* et al. (2020) available at `https://github.com/kohpangwei/group_DRO` as baseline code and integrate our approach into it. Most hyperparameter values are kept unchanged, and we list the important parameters along with model architectures for all the datasets in Table 18. For the simple models, we consider shallow 2D CNNs for the image datasets, and MLP and 1D CNN for the text data, as discussed in the previous section. In all cases, the simple model is trained for 20 epochs. For the final model, following Sagawa* et al. (2020), we use the Pytorch `torchvision` implementation of ResNet50 (He et al., 2016) with pre-trained weights on ImageNet data

for the image datasets, and the Hugging Face `pytorch-transformers` implementation of the BERT `bert-base-uncased` model, with pre-trained weights (Devlin et al., 2019) for the language-based datasets.

| Dataset | Simple Model | Optimizer | LR | BS | $\lambda_2$ | Final Model | Optimizer | LR | BS | $\lambda_2$ | $\lambda_c$ | $S$ | # epochs |
|---------|--------------|-----------|-----|-----|------------|-------------|-----------|-----|-----|------------|-------------|-----|----------|
| Waterbirds | 2DConvNet1 | Adam | $10^{-5}$ | 32 | $10^{-4}$ | ResNet50 | SGD | $5\times10^{-4}$ | 128 | $10^{-4}$ | 20 | 4 | 100 |
| CelebA | 2DConvNet2 | Adam | $10^{-5}$ | 32 | $5\times10^{-4}$ | ResNet50 | SGD | $3\times10^{-4}$ | 128 | 0.001 | 10 | 5 | 50 |
| MultiNLI | 2MLP | Adam | 0.005 | 16 | $10^{-4}$ | BERT | AdamW | $5\times10^{-5}$ | 32 | 0 | 75 | 10 | 5 |
| CivilComments | 1DConvNet | Adam | $10^{-4}$ | 16 | $10^{-4}$ | BERT | AdamW | $10^{-5}$ | 32 | 0.001 | 25 | 10 | 10 |

Table 18: Training details for subgroup robustness datasets.

Table 19 shows the values of LR, $\lambda_c$ and $S$ we consider for tuning for the final model. Following Sagawa* et al. (2020), we keep $\lambda_2 = 0$ for MultiNLI. For the rest, we consider $\lambda_2 \in \{0.0001, 0.0005, 0.001\}$. For re-

| Dataset | LR | $\lambda_c$ | $S$ |
|---------|-----|-------------|-----|
| Waterbirds, CelebA | $[1, 5] \times 10^{-4}$ | $\{10, 15, 20, 25, 50, 75\}$ | $\{4, 5, 6, 8, 10\}$ |
| MultiNLI, CivilComments | $\{1, 2, 5\} \times 10^{-5}$ | $\{10, 25, 50, 75\}$ | $\{10\}$ |

Table 19: Values considered for tuning the hyperparameters for training the final model for the four subgroup robustness datasets.

sults, we choose the model with the best worst-group accuracy on the validation set. For comparison, we consider the values reported in Liu et al. (2021a).

### G.3.4 Further Discussion on Experimental Results

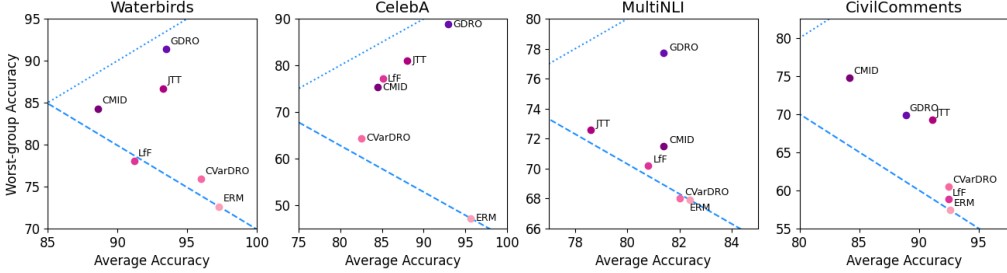

Figure 12: Scatter plots comparing worst-group and average accuracy of various methods on four benchmark datasets for subgroup robustness. The dotted line is the $x = y$ line. The dashed line represents the trade-off between average and worst-group accuracy.

Fig. 12 shows scatter plots comparing the worst-group and average accuracy of various methods on four benchmark datasets for subgroup robustness. The dashed line represents the trade-off between average and worst-group accuracy. Points on or close to this line correspond to methods that shift the balance between the invariant/spurious features so that the gains in the worst-group accuracy match the decrease in the average accuracy. We observe that GDRO in all cases as well as CMID and JTT in several cases (except Waterbirds and MultiNLI, respectively) lie far from this line. This suggests that in most cases, these methods allow for more gains in the worst-group accuracy compared to the decrease in the average accuracy.

### G.4 Bias in Occupation Prediction Experiment

We use a version of the Bios data shared by the authors of (De-Arteaga et al., 2019). We used the official implementation of (Cheng et al., 2023), available at `https://github.com/pinkvelvet9/snobpaper`, to obtain results and for comparison purposes. In this implementation, they consider 25 occupations and train separate one-vs-all linear classifiers for each occupation based on word embeddings to make predictions. We directly used their implementation to obtain results for ERM and Decoupled (Dwork et al., 2018) on the data.

For our approach, we employed linear models for both the simple model and the final model. We directly regularized the CMI with respect to the ERM from their implementation. The final model was trained for 5 epochs using SGD with LR $= 0.1, BS = 128, \lambda_c = 5, S = 5$. We only tuned the LR for this case, considering values of 0.05 and 0.1.

