# OpenReview forum: "Mitigating Simplicity Bias in Deep Learning for Improved OOD Generalization and Robustness"
_TMLR — Accepted by TMLR_

### Review · Reviewer_ojhn · 2024-05-13

**Summary Of Contributions:**

This paper proposes an assumption about simplicity bias: surrogate features are simple features that are still predictive of the label. This paper proposes a new framework that utilizes conditional mutual information regularization to alleviate simplicity bias, and theoretically analyzes its effectiveness. Experiments verify that the proposed framework can effectively alleviate simplicity bias and improve the model's generalization ability, subgroup robustness, and fairness.

**Audience:**

Yes

**Claims And Evidence:**

Yes

**Requested Changes:**

see above

**Strengths And Weaknesses:**

Strengths:
1.	This paper proposes a universal framework that can effectively alleviate the simplicity bias of neural network, and the proposed framework can be simply inserted into existing methods and improve their generalization ability, subgroup robustness, and fairness.
2.	This paper provides a comprehensive theoretical proof that its proposed framework can alleviate simplicity bias and enhance the generalization of neural networks on out of distribution data.
3.	This paper conducts experiments on datasets with different focuses in multiple fields and verifies the competitiveness and superiority of its proposed framework.
Weaknesses:
1.	The two-stage training strategy proposed in this paper first trains a simple model on the training data, and then trains a complex model by adding conditional mutual information terms to the loss function. However, the complexity of the models used in both theoretical and empirical analyses is the same. This causes inconsistency in the contextual statements.
2.	The paper proposes a universal framework for alleviating simplicity bias, but its experimental setup is relatively simple, and the experimental results do not reflect whether the framework is still effective when combined with existing state-of-the-art methods.
3.	There are some problems in the usage of symbols, which reduces its readability.

---

> ### Author Response · Authors · 2024-06-18
>
> We thank the reviewer for the valuable comments and suggestions.
>
> **W1: The complexity of the models used in both theoretical and empirical analyses is the same.**
>
> In Section 2, we define the complexity of a model in terms of its size (number of trainable parameters, layers, etc.). Based on this definition, specifically considering the number of trainable parameters, in both our theoretical and empirical analyses, the simple and complex models have different complexities.
>
> **W2: The experimental results do not reflect whether the framework is still effective when combined with existing state-of-the-art methods.**
>
> We thank the reviewer for the suggestion. While this is an interesting thought, our work introduces CMID as a standalone method, which is effective in mitigating simplicity bias and improving OOD generalization, robustness, and fairness across various dataset domains and modalities. We believe that exploring the effectiveness of CMID when combined with other methods is beyond the scope of this work, and hence we do not make any claims about the effectiveness of such a combination of CMID with existing state-of-the-art methods. We believe that combining various robustness techniques can be an interesting direction for future work.
>
> **W3: Problems in the usage of symbols.**
>
> We welcome suggestions that the reviewer may have about how we can improve readability and presentation. We would be more than happy to make modifications accordingly.

---

### Review · Reviewer_2xRY · 2024-05-29

**Summary Of Contributions:**

This paper first analyzes the hypothesis put forth that simple features are often spurious and lead to shortcut learning hurting the generalization to few subgroups. To counter this limitation, the paper proposes CMID, where they train a simple neural network first to learn the spurious features, and then train another (more complex?) network to "stay away" from learning these features using an MI minimization constraint. They show theoretical support for their hypothesis and better empirical evidence on many standard datasets.

**Audience:**

Yes

**Broader Impact Concerns:**

None.

**Claims And Evidence:**

Yes

**Requested Changes:**

1) More experimental results with advanced backbones.

2) Comparisons with zero shot methods like CLIP using comparable backbones.

3) Comparisons using distillation approaches.

**Strengths And Weaknesses:**

Strengths
-----------------

1. A simple method for improving robustness to spurious features with no group annotation required.
2. Adequate theoretical and empirical support provided to justify the hypothesis.
3. Better worst-group accuracy than prior methods.

Weaknesses
--------------------

1. The choice of "simpler" models vs "complex" models seems to be very task-specific and undecidable without access to a validation set with worst-group labels. Since this is fundamental to the problem, a more detailed suggestion on this front could help.

2. In some sense, this seems to be the reverse of the distillation problem - while in distillation, task-specific knowledge is "distilled" from a complex model to a simple model, here task-unrelated information is being avoided from a small model to a large model. On this note, more experimental results have to be presented comparing the accuracy using distillation from large models.

3. All the image datasets use CNNs, but ViTs have been more commonplace for a while now. The experiments have to incorporate the equivalent choice of "simpler" vs complex models when using ViTs. Similarly, results using training-free, zero-shot methods like CLIP have to be included in the comparisons.

4. While the worst-group accuracy is improved, this comes at the cost of average accuracy/dominant group accuracy - this trade off is more severe using CMID compared to prior approaches in table 8.

---

> ### Author Response · Authors · 2024-06-18
>
> We thank the reviewer for the helpful comments and feedback which have helped improve the paper.
>
> **W1: A more detailed suggestion on selecting simple and complex models could be helpful.**
>
> We agree with the reviewer that the methodology to select the simple model, as discussed in Section 3, is task-specific. In Section 3 and Appendix A.2, we also include experiments showing that the choice of smaller model architecture (within some range) does not significantly impact performance. Additionally, our method has a tunable regularization strength parameter. Therefore, as long as the simple model relies more on the spurious features as compared to other features, our approach to regularize CMI with respect to this model seems effective. On many of the tasks we consider, even the complex model primarily relies on the spurious features, due to simplicity bias in deep learning, so any simpler models should suffice.
>
>
> **W1: Model selection requires access to a validation set with worst-group labels.**
>
> We agree with the reviewer that access to a validation set with group labels is helpful, and sometimes necessary, for choosing the simple model class and tuning the regularization strength parameter. This is a limitation of our approach and several other approaches in this line of work [1, 2, 3], which we hope future work can overcome.
>
>
> **W2: Comparison with distillation approaches.**
>
> We appreciate the reviewer’s observation that the problem of mitigating simplicity bias and our approach are in some sense the reverse of distillation approaches. Based on this relation, we suspect that distillation approaches would not be effective in mitigating simplicity bias or improving OOD generalization. Even complex models display poor OOD generalization for the benchmarks considered in the paper due to simplicity bias in deep learning, and using smaller models to mimic their behaviors could further increase the reliance on simple or spurious features due to the limited capacity of smaller models. This suggests that distillation approaches could amplify the simplicity bias and further reduce the OOD performance. While directly comparing our method to distillation approaches may not be very informative, we have incorporated some discussion on the distillation approaches and their connection with the objective of mitigating simplicity bias in Appendix B. We believe that this can be useful for the reader.
>
> **W3: Comparisons with training-free zero-shot methods like CLIP.**
>
> We thank the reviewer for raising this important point of comparison. Below, we show the test performance of a CLIP ViT-B/16 model [4], for two datasets. To classify a test image, we first provide CLIP with text descriptions for each class. For waterbirds, we use [‘a landbird’, ‘a waterbird’], and for CelebA, we use [‘a non-blonde’, ‘a blonde’]. We then feed the test image into CLIP and choose the class with the highest cosine similarity to its output as the predicted label.
>
> | Dataset | Method | Average Accuracy | Worst-group Accuracy |
> |---|---|---|---|
> |Waterbirds | CLIP | 78.3 | 29.3|
> |     | CMID | 88.6 | 84.3 |
> | CelebA | CLIP | 88.7 | 40.4 |
> |      | CMID | 84.5 | 75.3 |
>
>
> We note that some of the benchmarks used in our paper are challenging due to imbalances in training data. For example, predicting “blonde” vs. “not-blonde” on the CelebA dataset is difficult because there are very few blonde males in the training data. The imbalances in CLIP training data, on the other hand, are unclear. While CLIP models may perform well on some benchmarks, it remains uncertain if they are effective in all cases. The results above demonstrate that CLIP models can be susceptible to natural spurious correlations, highlighting the need for grounded and consistent robustness techniques, such as the proposed method CMID. We have added the results and discussion in Appendix E.4 in the paper.
>
> **W3: Experiments using CMID with ViTs.**
>
> We agree with the reviewer that ViTs are a widely used architecture and it is important to verify that our method remains effective when applied to them. Below, we show test accuracies on the Waterbirds dataset when training a ViT-B/16 model [4] using our method compared to ERM. Due to limited time, we have yet to conduct full hyperparameter sweeps for CMID. However, these results still show that our method improves over ERM irrespective of the (complex) model architecture.
>
> | Dataset | Method | Average Accuracy | Worst-group Accuracy |
> |---|---|---|---|
> |Waterbirds | ERM | 83.8 | 67.3 |
> | | CMID | 84.8 | 70.1 |
>
> For our approach, we use the same shallow NN as the simple model class, as used in our subgroup robustness experiments in Section 5.3. We note that when considering our definition of model complexity (number of trainable parameters, layers, etc.), deep ViTs and CNNs are of the same scale. As a result, the choice of the simple model class can remain the same when considering the two architectures.

---

> > ### Author Response · Authors · 2024-06-18
> >
> > **W3 (contd.)**
> >
> > We have added these results in Appendix E.3 in the paper. We are happy to build upon these preliminary results should the reviewers find this helpful.
> >
> >
> > **W4: Tradeoff between ID and OOD accuracy in Table 8.**
> >
> > We appreciate the reviewer’s comment. As discussed in Appendix D, the use of methods that improve OOD generalization often leads to a drop in the ID accuracy. This is because spurious features are often more predictive on the IID test set compared to other features that may be more informative for OOD generalization. For the datasets considered in Table 8, we study this tradeoff between ID and OOD accuracy in more detail in Appendix G.3.4, Fig. 12. We see that most approaches allow for more gains in worst-group accuracy compared to the reduction in average accuracy. Moreover, CMID (as well as JTT and GDRO) seems better than other methods in terms of the tradeoff.
> >
> > **References:**
> >
> > [1] Jun-Hyun Bae, Inchul Choi, and Minho Lee. BLOOD: Bi-level learning framework for out-of-distribution generalization, 2022.
> >
> > [2] Evan Z Liu, Behzad Haghgoo, Annie S Chen, Aditi Raghunathan, Pang Wei Koh, Shiori Sagawa, Percy Liang, and Chelsea Finn. Just train twice: Improving group robustness without training group information. In Marina Meila and Tong Zhang, editors, Proceedings of the 38th International Conference on Machine Learning, volume 139 of Proceedings of Machine Learning Research, pages 6781–6792. PMLR, 18–24 Jul 2021.
> >
> > [3] Shiori Sagawa*, Pang Wei Koh*, Tatsunori B. Hashimoto, and Percy Liang. Distributionally robust neural networks. In International Conference on Learning Representations, 2020.
> >
> > [4] Alexey Dosovitskiy, et al. An image is worth 16x16 words: Transformers for image recognition at scale. arXiv:2010.11929, 2020.

---

### Review · Reviewer_nfY4 · 2024-06-05

**Summary Of Contributions:**

This paper addresses the issue of out-of-distribution (OOD) generalization in neural networks by building on concepts from previous studies on simplicity bias. Neural networks, even when overparameterized, tend to learn "simple" features, an empirically proven phenomenon known as simplicity bias. While simplicity bias enhances in-distribution generalization (evidenced by phenomena such as the double-descent curve), it undermines OOD generalization because simple features are often spurious (i.e., correlated with irrelevant parts of the input, such as the background). Consequently, learning more complex features that achieve similar in-domain classification accuracy but significantly better OOD classification performance is desirable.

The authors correctly note that, due to the No Free Lunch (NFL) theorem, there are no universal solutions to this problem, and distinguishing between invariant and spurious features requires some assumptions. They propose that simple (and hence spurious) features are those that can be learned with a "simple" neural network. Their method involves first training a simple neural network to perform the task, then training a more complex neural network architecture such that it i) achieves low training/validation error, and ii) learns features with low conditional mutual information with those of the simple neural network. They refer to this method as CMID.

Through several experiments, the authors demonstrate that CMID achieves superior OOD generalization compared to competing methods on well-known datasets containing both relevant (invariant) and irrelevant (spurious) features. They also provide some theoretical guarantees using a toy example of Gaussian mixture model data (with one spurious feature/dimension) and within a causal learning framework.

Overall, this paper is a strong experimental study (with some limited theoretical results), and its core idea is novel and beautiful. I have a number of questions and requests (see the Weaknesses section), and assuming these are addressed satisfactorily, I believe this paper is a good candidate for TMLR.

**Audience:**

Yes

**Broader Impact Concerns:**

-

**Claims And Evidence:**

Yes

**Requested Changes:**

- Please adjust the tables at the bottom of each page instead of in the middle of the text.

- Please see (weaknesses) section.

**Strengths And Weaknesses:**

**Strenghts**:

- The paper is well-written.

- The problem considered in this work is of interest to a wide range of audiences in the machine learning community, including researchers focusing on simplicity bias in neural networks, the role of information theory in machine learning, and those working on mitigating the effect of spurious correlations in ML models.

- The overall method is logical, natural, and appears novel. The idea is beautiful and holds significant potential for future development.

- Experimentations and additional materials in the Appendix section are thorough and comprehensive. Also, the method has been empirically shown to be quite effective in at least some real-world applications.

-------------------------

**Weaknesses**:

- The theoretical setting considered is very limited, but this might not be a weakness since the paper is primarily experimental.

- (**Major issue 1**) As far as I have noticed, the authors have not empirically investigated in-domain generalization. Training a complex model to deviate from a simpler one is intuitively justified to improve OOD generalization, but it may degrade in-domain generalization, especially when data is scarce. As the data volume increases, this issue is expected to diminish, but at what point does this phenomenon shift? How many more samples does your method require to achieve comparable in-domain generalization? Please either justify your method based on this argument or conduct additional experiments to address this issue.

- (**Major issue 2**) The simple model can learn "some" of the simple and spurious features, and through your CMI regularizer, these features do not transfer to the complex model. However, what if there are several spurious features? How large or small does the simple model need to be to i) only learn the simple and "bad" features, and ii) learn "all" such features? I believe this question is also left unanswered in the paper.

- (**Question**) In Table 7, both CMID and EIIL have a higher OOD accuracy compared to training accuracy. Why?

---

> ### Author Response · Authors · 2024-06-18
>
> We thank the reviewer for the positive feedback and helpful suggestions.
>
> **Major issue 1: As far as I have noticed, the authors have not empirically investigated in-domain generalization.**
>
> We appreciate the reviewer’s comment. In addition to OOD/worst-group test performance, we include the IID test performance for CMNIST and CPMNIST in Table 5, as well as the average (group-balanced) test accuracy on the subgroup robustness benchmark datasets in Table 8.
>
> We agree with the reviewer that the use of methods that improve OOD generalization often leads to a drop in the ID accuracy. Specifically, all methods with good OOD test accuracy on CMNIST and CPMNIST have lower ID test accuracy compared to ERM. Similarly, we see a tradeoff between the average and worst-group accuracies in Table 8, which we analyze in more detail in Appendix G.3.4, Fig. 12. This is because spurious features are often more predictive on the IID test set compared to other features that may be more informative for OOD generalization. We discuss this limitation in more detail in Appendix D.
>
>
> **Major issue 2: What if there are multiple spurious features?**
>
> We agree with the reviewer that this is an important case to consider. In Section 5.2, we consider the color+patch MNIST dataset, which has 2 spurious features, to illustrate the effectiveness of our approach even in the presence of multiple spurious features. Similarly, we consider the Camelyon-17 dataset, where it is unclear how many spurious or “bad” features are present, which make OOD generalization difficult. On both datasets, we select the simple model class using the methodology described in Section 3, and see that our approach is effective.
>
> One challenging setting could be the presence of multiple spurious features with varying levels of complexity. Such a situation may require multiple rounds of applying our approach: training a simpler model and using it to regularize the complex model. However, we have not come across any datasets that may warrant the use of multiple rounds of debiasing, except the use of CPMNIST in [1], which we also consider in our work. We welcome any further suggestions the reviewer may have on assessing the method’s effectiveness in the presence of multiple spurious features.
>
>
> **Question: Why is training accuracy lower than OOD test accuracy for EIIL and CMID in Table 7?**
>
> We thank the reviewer for this question. Both these methods are designed to improve OOD generalization, and the best model is selected based on the performance on the validation set. This can be a reason why the OOD test accuracy is better than the training accuracy. If the user wants to ensure good training performance, they can change the regularization strength or the model selection rule to select the model by taking both train and validation set accuracies into account. We discuss this limitation of how methods designed for OOD generalization can lead to lower ID generalization in Appendix D. We have added some of this discussion at the end of Appendix D.
>
>
> **References**
>
> [1] Jun-Hyun Bae, Inchul Choi, and Minho Lee. BLOOD: Bi-level learning framework for out-of-distribution generalization, 2022. URL https://openreview.net/forum?id=Cm08egNmrl3

---

> > ### Comment · Reviewer_nfY4 · 2024-06-25
> > **Response to author(s)**
> >
> > Thanks for the response. The majority of my concerns have been obviated.

---

### Comment · Action_Editor_c42s · 2024-06-15
**Discussion period**

Now that we have 3 reviews submitted, we need to start the discussion period.

Dear authors, please take a look at the reviews and submit your responses or update the paper if necessary.

Dear reviewers, thank you so much for submitting the reviews! Please take a look at the other reviews, and engage in a discussion with authors when they respond.

Thank you,
AE

---

### Author Response · Authors · 2024-06-18
**General Response**

We thank the reviewers for their time and effort to review our work and for providing valuable comments and feedback to help improve the paper. We are encouraged that the reviewers find our work well-written and of interest to a wide range of audiences that span the ML community (reviewer nfY4), and our proposed approach logical, novel (reviewer nfY4), simple (reviewer 2xRY), effective in real-world applications (reviewer nfY4), improving robustness to spurious features without group annotations (reviewer 2xRY), and easily incorporable into existing methods (reviewer ojhn). We are pleased they acknowledge our method’s thorough and comprehensive empirical evaluation (reviewer nfY4) showing its competitiveness and superiority over prior approaches (reviewers 2xRY, ojhn), and our theoretical results showing it can alleviate simplicity bias and enhance the OOD generalization (reviewer ojhn).

We have tried our best to address the reviewers’ concerns in the responses to each reviewer. We have updated the paper with the following modifications:

- We have added some discussion on how the regularization strength parameter for CMID can help reduce the tradeoff between ID and OOD accuracy at the end of Appendix D, in response to reviewer nfY4’s question.

- We have moved the tables in the main body to the top or bottom of each page, as per the suggestion of reviewer nfY4.

- Based on reviewer 2xRY’s observation that mitigating simplicity bias is in some sense the reverse of distillation, we have added discussion on this connection in Appendix B, under the ‘knowledge distillation’ paragraph.

- As suggested by reviewer 2xRY, we have added results for our approach when using ViT, and comparison with CLIP in Appendix E.3 and E.4, respectively.

---

### Decision · Action_Editor_c42s · 2024-08-04

**Recommendation:** Accept as is

**Comment:**

I believe, there are several downsides to the paper:

1. The core intuition that the spurious features correspond to simple features that correlate with the target is not novel, and is in fact the default understanding of the spurious correlation issue. This point may be taken as subjective.

2. There are counter-examples to the above intuition, at least if taken literally. The authors discuss how the Celeb-A dataset doesn't follow this intuition. Moreover, [this paper in Appendix B.1](https://arxiv.org/abs/2204.02937) shows that when trained on the *inverse waterbirds problem*, i.e. when the label is associated with the background and the bird is a spurious feature, the models still perform poorly on minority groups. In other words, the poor worst group performance on waterbirds cannot be explained by the background being simpler than foreground.

3. The proposed method is qualitatively very closely related to multiple existing methods. [JTT](https://arxiv.org/abs/2107.09044) also trains  two models, where the first one is supposed to learn the spurious features, and the second one uses predictions from the first one. There are also many other very similar methods: [LfF](https://arxiv.org/abs/2007.02561), [AFR](https://arxiv.org/abs/2306.11074), etc.

4. The results reported are not outstanding, compared to other recent methods. For example, [AFR](https://arxiv.org/abs/2306.11074) achieves stronger results on waterbirds, CelebA and MultiNLI, as far as I can tell.

Despite these limitations, I believe this paper meets the standards of TMLR, and will be a valuable contribution to the journal. I think the paper should be accepted.

Minor: I would recommend that the authors include stronger more recent baselines in Table 8: e.g. [AFR](https://arxiv.org/abs/2306.11074), [CnC](https://arxiv.org/abs/2203.01517).

**Audience:**

The topic of out-of-distribution generalization is central to machine learning, and the paper will be of interest to TMLR community.

**Claims And Evidence:**

The claims made in the paper are supported by evidence. Specifically:
- The paper makes a hypothesis that spurious features correspond to simple features that correlate with the label.
- The authors propose a simple method based on (1) training a simple model on the data and (2) regularizing the main model to have low mutual information in predictions with the simple model.
- There are theoretical results showing that the method learns the invariant features in a toy model.
- The authors show empirical results across a broad range of tasks: simplicity bias, subgroup robustness, fairness.

The appendix presents additional details, results and ablations.